# Evaluation of the dual-polarization weather radar quantitative precipitation estimation using long-term datasets

Tanel Voormansik[1,2], Roberto Cremonini[3,4], Piia Post[1], Dmitri Moisseev[4,5]

[1] Institute of Physics, University of Tartu, Estonia

[2] Estonian Environment Agency, Estonia

[3] Regional Agency for Environmental Protection of Piemonte, Department for Natural and Environmental Risks, Torino, Italy

[4] Institute for Atmospheric and Earth System Research / Physics, University of Helsinki, Finland

[5] Finnish Meteorological Institute, Helsinki, Finland

*Correspondence*: Tanel Voormansik (tanel.voormansik@ut.ee)

**Abstract.** Accurate, timely and reliable precipitation observations are mandatory for hydrological forecast and early warning systems. In the case of convective precipitation, traditional rain gauge networks often miss precipitation maxima, due to density limitations and high spatial variability of rainfall field. Despite several limitations like attenuation or partial beam-blockings, the use of C-band weather radar has become operational in most European weather services. Traditionally, weather radar-based quantitative precipitation estimation (QPE) is derived from horizontal reflectivity data. Nevertheless, dual-polarization weather radar can overcome several shortcomings of the conventional horizontal reflectivity based estimation. As weather radar archives are growing they are becoming increasingly important for climatological purposes in addition to operational use. For the first time, the present study analyses one of the longest datasets from fully operational polarimetric C-band weather radars; those are located in Estonia and Italy, in very different climate conditions and environments. The length of the datasets used in the study is 5 years for both Estonia and Italy. The study focuses on long-term observations of summertime precipitation and their quantitative estimations by polarimetric observations. From such derived QPEs accumulations for 1 hour, 24 hours, and one month durations are calculated and compared with reference rain gauges to quantify uncertainties and evaluate performances. Overall, the radar products showed similar results in Estonia and Italy when compared to each other. The product where radar reflectivity and specific differential phase were combined based on a threshold exhibited the best agreement with gauge values on all accumulation periods. In both countries reflectivity based rainfall QPE underestimated and specific differential phase based product overestimated gauge measurements.

## 1. Introduction

Detailed surface rainfall information is of great importance in many fields not only for agricultural or hydrological applications. In the recent past the COST 717 Action entitled "Use of Radar Observations in Hydrological and NWP models" investigated the assimilation of weather radar based precipitation in NWP (Macpherson, 2004). Weather radar data have been assimilated in a variety of assimilation systems and models of increasing resolution. At the beginning the latent heat nudging was the most popular technique (Gregorč et al., 2000), while researchers recently moved towards volume reflectivity assimilation techniques: for example, Schraff et al. (2016) proposed the KENDA (ensemble Kalman filter for convective-scale data assimilation) operator to assimilate reflectivity volume data in the COSMO (COnsortium for Small-scale MOdelling) model. For decades gauge networks have provided the best reference datasets. E-OBS 50-years daily European gridded interpolated dataset has been widely used in climatological studies (Cornes et al., 2018). Gauge-based datasets have well-known shortcomings in their low spatial and to a lesser degree temporal resolution. Precipitation data from satellites provides good spatial coverage, but still not in very high temporal resolution, especially in higher latitudes (Sun et al., 2018). Polar-orbiting satellites provide better spatial resolution data in higher latitudes, but they are very limited in temporal resolution (Tapiador et al., 2018). What is more, satellite-based precipitation estimates are limited by the accuracy of the estimates. The accuracy of the estimates has a regional dependency and therefore can vary due to the physiography of the study areas (e.g. precipitation climate, land use, and geomorphology) (Petropoulos and Islam, 2017). Now that weather radars have been used already for decades in many countries their archives are getting long enough to use the data in climate studies (Saltikoff et al., 2019). In

the last decade, various studies have used multi-year single-polarization weather radar data successfully in deriving rainfall climatology with high spatiotemporal resolution (Overeem et al., 2009; Goudenhoofdt et al., 2016). However, quantitative precipitation estimation (QPE) with single-polarization C-band radar is strongly affected by attenuation of the electromagnetic wave in heavy precipitation or a wet radome, hail contamination, partial beam blockage, and absolute radar calibration (Krajewski et al., 2010; Cifelli et al., 2011).

All prior shortcomings can be mitigated by the use of dual-polarization weather radar data. Several studies have shown that rainfall retrieved from dual polarimetric radar differential phase measurements outperforms rainfall estimated from horizontal reflectivity, especially in heavy precipitation (Wang and Chandrasekar, 2009; Vulpiani et al., 2012; Wang et al., 2013; Crisologo et al., 2014). Because differential phase measurements tend to be noisy and less reliable in low-intensity precipitation Crisologo et al. (2014) and Vulpiani and Baldini (2013) improved the robustness of their rainfall retrieval technique by employing a combination of horizontal radar reflectivity $R(Z_H)$ and specific differential phase $R(K_{DP})$ where a threshold was set below which $R(Z_H)$ was used and over which $R(K_{DP})$ was used. Bringi et al. (2011) also compared performances of $R(Z_H)$, $R(K_{DP})$, and the combination product of the two on a relatively long set of data of four years.

The main aim of this study is to evaluate the potential of using polarimetric weather radar QPE on long-term warm-season datasets in various climatological environments. Previous studies where the benefits of dual polarimetric radar QPE have been shown are mostly based on selected short periods or only single events (Wang and Chandrasekar, 2010; Chang et al., 2016; Montopoli et al., 2017; Cao et al., 2018). While the performance of the QPE methods can be compared based on short periods as well, only a study based on long-term data can prove the robustness of a method and suitability for long-term operational use. The uniqueness of this paper is ensured by various features. First of all, we have a long 5-year dataset, starting already from 2011, derived by operational dual polarimetric C-band weather radar made by different manufacturers. The dataset is gathered from the archive of weather radar scans set up for operational surveillance in the meteorological services. Secondly, the study areas are from heterogeneous climatologies being the weather radar located in Estonia and Italy. This is also the first-ever study evaluating weather radar QPE in Estonia. What is more, we will assess the effect of radar scan interval as the radar data scan frequency is 5 and 15 minutes from Italy and Estonia respectively. The study analyses results first in a few selected cases. The whole dataset is analysed at three accumulation intervals of 1 hour, 24 hours, and one month. Three radar QPE products are generated for comparison. First the horizontal reflectivity based product $R(Z_H)$, then the specific differential phase based product $R(K_{DP})$ and as a third radar QPE product, an $R(Z_H)$ and $R(K_{DP})$ combination. To investigate the performance of all these weather radar based QPE products they are compared with gauge accumulations.

The paper is organized as follows. Section 2 describes the rainfall estimation datasets from radar and rain gauges and methods used for comparisons. The results are discussed in Sect. 3. In Sect. 4 conclusions are provided.

## 2. Data and methods

### 2.1 Statistical methods for comparison

To estimate the performance of the radar rainfall products they were compared with gauge accumulations. The study period was limited to the warm season (May - September for Estonia and April - October for Italy). In Estonia, the mean annual precipitation is 649 mm. Precipitation climatology has distinct seasonality with maximum in summer (215 mm) followed by autumn (198 mm), winter (128 mm), and spring (108 mm). The summer maximum of seasonal mean precipitation is especially pronounced in the continental part of Estonia (246 mm in Mauri, South-East Estonia), Tammets et al. (2013).

In Piemonte, close to the radar, the mean annual precipitation is 870 mm having a bimodal distribution with peaks in spring (266 mm) and autumn (255 mm), Devoli et al. (2018).

Radar-based QPEs have been accumulated to the 1-hour duration and longer durations have been calculated based on these accumulations. Accumulations were calculated by adding subsequent instantaneous radar QPE values without any space-time interpolation. No missing data for radar or gauges was tolerated to prevent underestimation. A threshold of 0.1 mm was set and applied such that both gauge and radar QPE values must exceed this value to make the pair valid.

The quality of the rainfall estimates was estimated by the following verification measures (where $r_i$ is the $i$-th out of $n$ radar precipitation estimates, $g_i$ the $i$-th out of $n$ gauge observations, $r_m$ the mean of all $n$ radar precipitation estimates, and $g_m$ the mean of all $n$ gauge observations):

Pearson's correlation coefficient: $CC = \frac{\sum_{i=1}^{n}(r_i - r_m) \cdot (g_i - g_m)}{\sqrt{\sum_{i=1}^{n}(r_i - r_m)^2} \cdot \sqrt{\sum_{i=1}^{n}(g_i - g_m)^2}}$, (1)

Normalized Mean Absolute Error: $NMAE = \frac{\sum_{i=1}^{n}|r_i - g_i|}{\sum_{i=1}^{n} g_i}$, (2)

Normalized Mean Bias: $NMB = \frac{\sum_{i=1}^{n}(r_i - g_i)}{\sum_{i=1}^{n} g_i}$, (3)

Root Mean Squared Error: $RMSE = \sqrt{\frac{1}{n}\sum_{i=1}^{n}(r_i - g_i)^2}$, (4)

Nash-Sutcliffe Efficiency: $NASH = 1 - \frac{\sum_{i=1}^{n}(r_i - g_i)^2}{\sum_{i=1}^{n}(g_i - g_m)^2}$. (5)

The Nash coefficient is typically used to assess the accuracy of hydrological predictions, but it has also been used for weather radar-based rain rates and gauge comparisons (Nash and Sutcliffe, 1970).

## 2.2 Rain gauge measurements

In Estonia major renewal and automation of the rain gauge network run by the Estonian Environment Agency (EstEA) started in 2003. From 2003 to 2006 the network was updated to automatic tipping-bucket gauges. Starting from 2006 the tipping-bucket gauges were progressively replaced by weighted gauges. This process was finished by the end of the year 2011. By that time there were 33 automatic weighted gauge stations and 27 stations with tipping-bucket gauges. According to the comparative study of parallel measurements of the tipping-bucket gauges and weighted gauges, the latter exhibited much higher quality (Alber et al., 2015). From the end of 2010, the data has been recorded with a 10-minute interval. Until 2010 the temporal resolution was one hour. Both 10-minutes and 1-hour data are being saved by EstEA since then, but only 1-hour data have been quality controlled by EstEA staff. Because the 10-minutes data are not quality controlled 1-hour gauge data was used in this study as a more reliable ground truth. The off-line manned data quality control includes using mainly weather sensor data as an additional source for comparisons but also neighbouring stations and weather radar data on some occasions. Only weighted gauge data was used because of the higher quality of these measurements and to ensure uniformness of the dataset. In this work 8 rain gauges close to Sürgavere, Estonia are included (Fig. 1). Data is with 0.1 mm resolution.

Since 1987, Arpa Piemonte, the regional agency for environment protection in Piemonte, Italy, operates a regional automatic gauges network made of about 380 tipping-bucket gauges. Most of the gauges are heated to avoid solid precipitation accumulation during the cold season. The temporal resolution of the gauges network is 1-minute. The Arpa Piemonte weather stations are equipped with CAE PMB2 tipping-bucket rain gauges. Their resolution (0.2 mm) is the amount of precipitation for one tip of the bucket. The working range of measures is from zero mm to 300 mm/h with underestimation for high precipitation intensities. Such errors are corrected according to results of WMO Field Intercomparison of Rainfall Intensity Gauges (Vuerich et al., 2009). Automatic data quality check is run on real-time data, followed by off-line manned data validation. In this study, a network subset made of 42 rain gauges close to Torino, Italy, has been considered (Fig. 1). Precipitation measurements range from 2012 to 2016.

## 2.3 Weather radar precipitation estimation

Data from C-band dual-polarization Doppler weather radars in Estonia and Italy were used in this study. The weather radars considered in this study are from different manufacturers, in Estonia Vaisala WRM200, and Italy Leonardo Germany Gmbh METEOR 700C radar. Figure 1 illustrates the location of Estonian radar (Sürgavere) and Italian radar (Bric della Croce) together with the locations of available rain gauges.

Sürgavere radar, located in central Estonia at altitude 128 m a.s.l., has been operational since May 2008 but for this study data

starting from 2011 was used because the gauge network was updated by that time. The radar performs a surveillance volume scan at 8 elevation angles (0.5°, 1.5°, 3.0°, 5.0°, 7.0°, 9.0°, 11.0°, and 15.0°) every 15 min starting each scan from the lowest elevation angle. Only the lowest elevation angle data were used. The resolution of the raw radar data is 300 m in range and 1° in azimuth. Data up to 10 km from radar were discarded because of the ground clutter and unreliable $K_{DP}$ estimation. Close to the radar stable and reliable differential phase observations are not available due to both the antenna itself and the TR-limiters

response time or the dual-pol switch in case of alternate transmission. Doppler filter was used to eliminate residual non-meteorological fixed clutter. In addition to speckle and clutter to signal ratio filtering at the signal processor level, polarimetric hydrometeor classification was used to filter non-meteorological targets from the display (Chandrasekar et al., 2013). After careful analysis, some of the data from Sürgavere radar had to be omitted completely. Years 2014 and 2015 were excluded because of gradually decreasing polarimetric data quality caused by a broken limiter which was replaced in March 2016. Data

from 2017 was discarded because the quality was inconsistent as a result of a broken stable local oscillator (STALO) which was replaced in May 2018. From Estonia, the investigated period ranges then from 2011-2018 and includes 5 years of data.

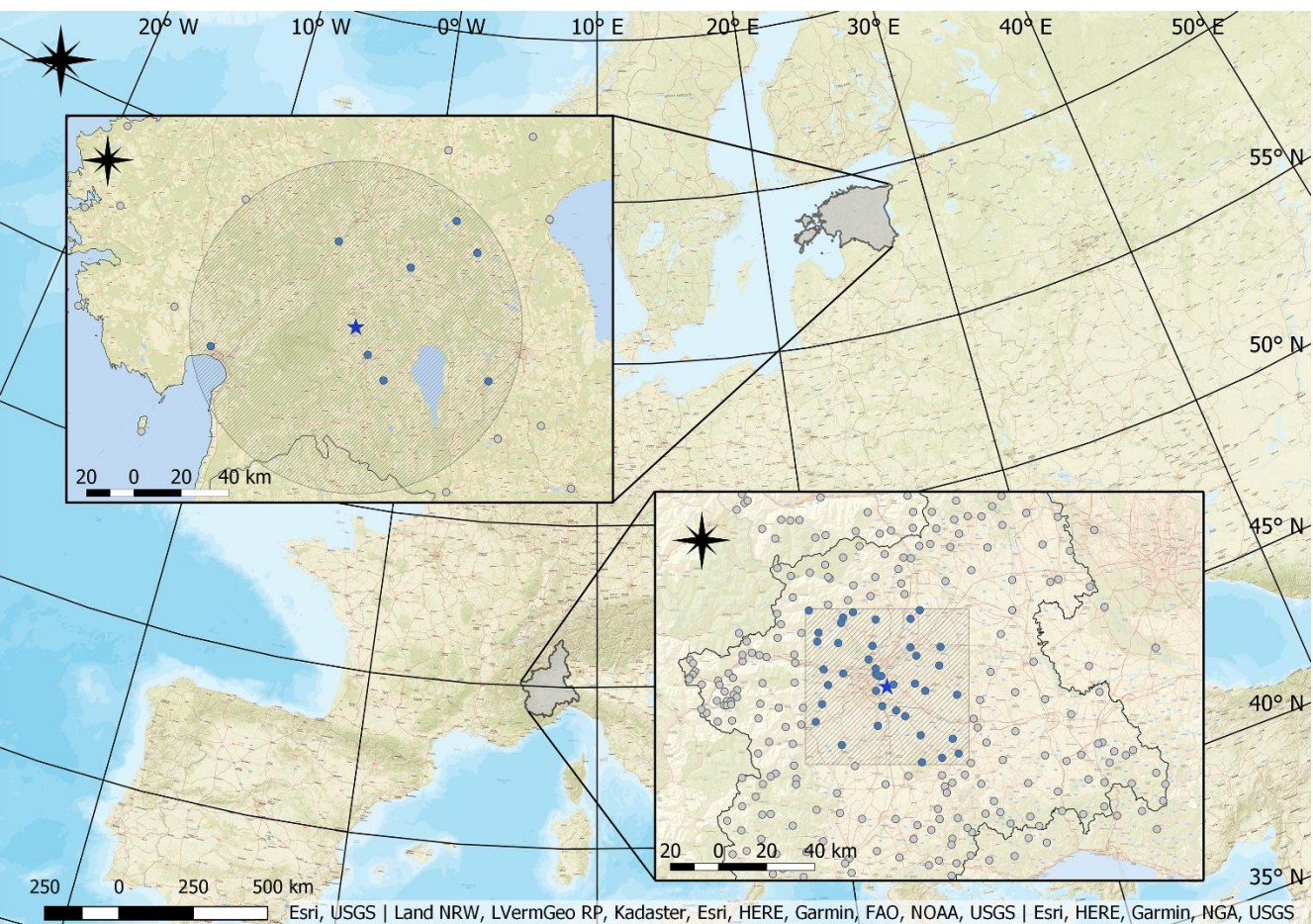

**Figure 1.** Study areas (shaded) located in Estonia (upper left zoomed area) and in Piemonte, Italy (lower right zoomed area). Grey dots denote gauge locations of Estonian and Piemonte region respectively and blue dots gauges inside the study area.

Blue stars reveal radar locations.

On the Torino hill, at altitude 770 m a.s.l., the operational dual-polarization Doppler C-band weather radar Bric della Croce is located. The radar site is in the central part of the Piemonte region: toward west and north at about 20 km the Alps start with peaks 2,500 - 3,000 meters above sea level. The radar performs fully polarimetric volume scans, made by eleven elevations up to 170 km range, with 340 meters range bin resolution. Bric della Croce observations used in the study ranged from 2012 to 2016 whereas observations from 2012 to 2013 are with ten-minutes interval and from 2013 to 2016 with five minutes interval. As can be seen from Fig. 1 circular area around the radar is used in Estonia but in Italy rectangular area is used. The reason for this is that orography in Piemonte is very complex ranging from flat plains in the Po valley (about 100 m a.s.l.) to the Alps up to more than 4,000 m a.s.l. The Bric della Croce weather radar is located on Torino hill that is about 30 km from the Alps. Therefore, the elegant and simple limitation in range by some kilometres from the radar site does not work. To avoid mountainous areas, where partial and total beam-blocking, heavy ground contamination increases, a rectangle area, that extends towards flat grounds, has been preferred.

The maximum distance of the gauges to be included in the comparison was limited to a 70 km radius from the radar location in the case of Estonia and up to 30 km distance in Italy. Thus, in Estonia and Italy rainfall data were from 8 and 42 gauges respectively. By limiting data analysis to warm season and constraining the maximum radar range, we were able to ensure that radar data were originating mainly from liquid precipitation (hail can also occur) which is required for more reliable rainfall intensity estimation. The possible occurrence of hail was not removed from the data because of the intention to keep additional data processing minimal and allow equal comparison of the various QPE methods. In the case of Italy, the applied range limit is also aimed at eliminating uncertainties due to complex orography, like shielding by the mountains, overshooting, bright band contamination.

QPEs, based on horizontal reflectivity, are extensively described by Cremonini and Bechini (2010) and by Cremonini and Tiranti (2018), meanwhile, $K_{DP}$ precipitation estimates are derived according to Wang et al. (2009). When $K_{DP}$ was equal to or less than zero, then $R(K_{DP})$ was set to zero. The area close to the weather radar up to eight kilometres has been left out due to heavy ground clutter contamination and unreliable estimations of $K_{DP}$.

Sürgavere radar specific differential phase ($K_{DP}$) and differential propagation phase ($\phi_{DP}$) were recalculated from raw $\phi_{DP}$ data using the Python ARM Radar Toolkit (Py-ART) (Helmus and Collis, 2016) function phase_proc_lp (Giangrande et al., 2013) with carefully tuned parameter values according to data specifics. With default parameter values the rays where differential propagation phase folding occurred did not unfold correctly and thus the function did not produce correct specific differential phase values. To fix the folding issue function parameters *self_const* (self-consistency factor) and *low_z* (the low limit for reflectivity – reflectivity below this value is set to this limit) had to be tuned. The self-consistency factor takes into account the spatial variability of reflectivity and differential reflectivity within a given path. It is used to improve $K_{DP}$ field behaviours to more closely follow the cell patterns found in $Z_H$. The default values for *self_const* and *low_z* were 60000.0 and 10.0 respectively and after testing with various combinations of various values the values 12000.0 and 0.0 were found to produce optimal results and therefore were chosen for final calculations. The values were first chosen after preliminary tests with single scans from multiple years between 2011-2018 and then confirmed after a final test with one month of 1-hour accumulation data from August 2018. The quality of the results was evaluated by using the verification measures introduced in Sect. 2.1 (Eq. 1-5). The final test results are shown in Table 1. The product with optimal values for the $K_{DP}$ processing algorithm ($R(K_{DP\ tuned})$) improves all verification measures when compared to the product based on the $K_{DP}$ processing with default parameters values ($R(K_{DP\ def})$). The $K_{DP}$ retrieval process involves filtering that reduces the range resolution of $K_{DP}$ to approximately 1 km.

Horizontal reflectivity ($Z_H$) was re-calibrated using a method that utilizes the knowledge that $Z_H$, $Z_{DR}$ (differential reflectivity), and $K_{DP}$ are self-consistent with one another and one can be computed from two of the others. $Z_{DR}$ is not suitable for QPE on C-band radars, but it can be used in this calibration methodology after applying strict restrictions on the data used for this purpose. The calibration was carried out using the self-consistency theory set down in Gorgucci et al. (1992), Gorgucci et al. (1999) and Gourley et al. (2009) where the methodology is described in detail. The method essentially compares the observed

differential propagation phase ($\phi_{DP}{}^{obs}$) to a calculated theoretical differential propagation phase ($\phi_{DP}{}^{th}$). The data used for calibration had to be filtered using several restrictions: only data from June to September was allowed; data from 0.5° elevation and 10-70 km range only used; only bins where horizontal and vertical polarization channel correlation coefficient was over 0.92 were used; any bins where $\phi_{DP}$ was greater than 12° were removed; whole ray where reflectivity was greater than 50 dBZ was removed; whole ray where $Z_{DR}$ was greater than 3.5 dB was rejected; only rays where $\Delta\phi_{DP}{}^{obs}$ was greater than 8° and where the consecutive rain path was at least 10 km were used; any scans in which precipitation occurred on top of the radome were removed. As a result, $Z_H$ bias values from the range of -2.0 to -5.0 dB were obtained depending on the date. The bias values were used to correct the corresponding observed $Z_H$ before to rain rate estimation. The impact of the re-calibration was evaluated on one month of 1-hour accumulation data from August 2018 using the verification measures introduced in Sect. 2.1 (Eq. 1-5). The verification results are presented in Table 1. QPE product based on re-calibrated reflectivity ($R(Z_{H\,cal})$) shows clearly superior results compared to the non-calibrated reflectivity based product ($R(Z_{H\,def})$), most notably by decreasing the negative bias.

To convert reflectivity $Z_H$ to rainfall rate R (mm/h) the following relation was used:

$$Z_H = 300R^{1.5}. \tag{6}$$

Specific differential phase $K_{DP}$ was converted to rainfall rate using the expression suggested by Leinonen et al. (2012):

$$R = 21.0K_{DP}^{0.720}. \tag{7}$$

The QPE of $R(Z_H)$ can be affected by attenuation on C-band radars especially in heavy precipitation and at long distances. While this can be corrected using $\phi_{DP}$ in our study it was not applied to the reflectivity data to not introduce another possible source of error between the results of Estonia and Italy that could not be easily quantified. The effectiveness of attenuation correction using $\phi_{DP}$ is hampered by its temperature, shape, and size distribution dependence which affect the accompanying error (Vulpiani et al., 2008). The QPE of $R(Z_H)$ can also be affected by the effect of the non-uniform vertical profile of reflectivity (VPR). In the current study, the effect of VPR will be limited because only data from the warm season was used and distance limits to the radar data were set (70 km for Estonia and 30 km for Italy, respectively).

Several studies have shown that $R(K_{DP})$ provides much more reliable intensity estimates in heavy rainfall (Vulpiani et al., 2012; Wang et al., 2013; Chen and Chandrasekar, 2015). On the other hand, it has been indicated that $K_{DP}$ retrieval itself is less reliable in light precipitation conditions (Giangrande and Ryzhkov, 2008; Ryzhkov et al., 2014). Thus, combining the two methods has the potential to be superior to using each method separately. For example, Vulpiani et al. (2013) used a weighted combination of $R(Z_H)$ and $R(K_{DP})$ where only reflectivity data was used for bins with $K_{DP}$ less than or equal to 0.5 °/km, and $K_{DP}$ was used additionally with increasing weight over that value up to 1 °/km over which it was solely used. Cifelli et al. (2011) used a simple threshold method where $R(K_{DP})$ was used when $R(Z_H)$ was exceeding 50 mm/h intensity. Several authors have successfully added $R(Z_{DR})$ based intensity estimation to the combination on S-band weather radars (e.g. Ryzhkov and Zrnic, 1995; Ryzhkov et al., 2005; Chandrasekar and Cifelli, 2012). Due to residual effects such as resonance, noise, and attenuation $R(Z_{DR})$ should not be used at C-band (Ryzhkov and Zrnic, 2019).

In our study rainfall from a combined threshold approach was used for both weather radars as a third product $R(Z_H,K_{DP})$. In the combined product $R(Z_H)$ was used in areas with $Z_H$ less than or equal to 25 dBZ and $R(K_{DP})$ otherwise if available. The $Z_H$ threshold value was selected after testing with various reflectivity levels. The reflectivity threshold was selected after verifying QPE performances at different reflectivity levels from 15 dBZ to 35 dBZ by 5 dBZ steps. The evaluation was based on 1-hour accumulation rainfall for August 2018 in Estonia and the verification statistics introduced in Sect. 2.1 (Eq. 1-5) were applied using the same gauges employed in the latter parts of the study as reference. From Table 1 it can be seen that the best scores are reached using 25 dBZ (QPE product $R(Z_{H25}, K_{DP})$). The same evaluation for the $R(Z_H,K_{DP})$ algorithm was carried out for Bric della Croce in Italy with a 1-hour accumulation period and it also confirmed the suitability of the 25 dBZ level. The

threshold level is considerably lower than some of the thresholds used in the literature referred to above but on our datasets it performed the best.

**Table 1**. Verification results of the test dataset of one month (August 2018) of the radar-based rainfall 1-hour accumulation products of Estonia. $Z_H$ not corrected for attenuation.

| | $R(Z_{H\,cal})$ | $R(Z_{H\,def})$ | $R(K_{DP\,tuned})$ | $R(K_{DP\,def})$ | $R(Z_{H15}, K_{DP})$ | $R(Z_{H20}, K_{DP})$ | $R(Z_{H25}, K_{DP})$ | $R(Z_{H30}, K_{DP})$ | $R(Z_{H35}, K_{DP})$ |
|---|---|---|---|---|---|---|---|---|---|
| CC | 0.699 | 0.699 | 0.659 | 0.428 | 0.687 | 0.721 | 0.726 | 0.713 | 0.705 |
| NMAE | 0.572 | 0.634 | 1.074 | 1.491 | 0.855 | 0.69 | 0.605 | 0.596 | 0.595 |
| NMB | -0.212 | -0.421 | 2.652 | 4.958 | 1.441 | 0.655 | 0.067 | -0.118 | -0.184 |
| RMSE (mm) | 1.611 | 1.709 | 2.329 | 2.656 | 2.071 | 1.832 | 1.714 | 1.718 | 1.704 |
| NASH | 0.247 | 0.202 | -0.088 | -0.241 | 0.032 | 0.144 | 0.199 | 0.197 | 0.204 |

The impact of the temporal sampling was analysed using Italian Bric della Croce weather radar second elevation PPI data which produces a 5-minute interval dataset. A degraded dataset of a length of one day, October 10th 2020, with a 15-minutes sampling rate was created by removing 2 out of 3 files. Hourly accumulation was calculated based on both sampling rates which resulted in a sample size of 253,514. As expected from the comparison of these accumulation pairs the obtained normalized mean bias was close to zero (0.03) while the correlation coefficient was 0.922 and the normalized mean absolute error 0.21.

If we compare different skill scores for 1-hour QPEs in Estonia and Italy, part of the differences in correlation coefficient and normalized mean absolute error can be explained as due to different time sampling. Table 2 below summarizes correlation coefficient and normalized mean absolute error in Estonia and Italy.

**Table 2.** Verification of the 1-hour accumulation QPE products of Estonia and Italy and differences without ("Difference") and with ("Comp. Diff") compensating the impact of the temporal sampling. CC and NMAE values are obtained from Table 3 and Table 4. $Z_H$ not corrected for attenuation.

| | $R(Z_H)$ | $R(K_{DP})$ | $R(Z_H, K_{DP})$ | | $R(Z_H)$ | $R(K_{DP})$ | $R(Z_H, K_{DP})$ |
|---|---|---|---|---|---|---|---|
| CC (Estonia) | 0.679 | 0.674 | 0.697 | NMAE (Estonia) | 0.537 | 0.868 | 0.594 |
| CC (Italy) | 0.843 | 0.808 | 0.870 | NMAE (Italy) | 0.531 | 0.514 | 0.423 |
| Difference | -0.164 | -0.134 | -0.173 | Difference | 0.006 | 0.354 | 0.171 |
| Comp. Diff. | -0.086 | -0.056 | -0.095 | Comp. Diff. | -0.204 | 0.144 | -0.039 |

Compensating the values obtained in Estonia for loss of correlation (0.078) and increased NMAE (0.21) due to 15-minutes time sampling with values estimated in Italy, it is visible that CC and NMAE are comparable in Estonia and Italy (last row in Table 2). It is worth noting that after the compensation is applied, QPE estimated by $R(Z_H)$ shows lower NMAE in Estonia.

The difference in NMAE of $R(K_{DP})$ and $R(Z_H)$ QPEs might stem from different precipitation regimes (more intense precipitation in Italy).

## 3. Results and discussion

### 3.1 Case comparisons

In this section radar QPE products are compared with single location gauge measurements of selected short periods from Estonia and Italy. This allows for evaluating the performance of the radar QPE against gauge measurements from a time-series viewpoint.

Figure 2 shows one month of precipitation on Jõgeva station location (60 km away from the radar site) in Estonia with 1-hour temporal resolution. Overall radar products follow the gauge measurements well but there are considerable differences among them. Reflectivity based product $R(Z_H)$ is not affected by noise and clutter in clear weather or light rain cases but on the other hand, it is underestimating rainfall amounts particularly in medium to heavy precipitation cases. By the end of the month, its sum of 40.5 mm was 19.6 mm less than gauge measured accumulation (70.1 mm). $R(K_{DP})$ then again is heavily overestimating precipitation amounts, especially during light rain cases. By the end of the month, the accumulated amount of 150.2 mm was more than double the gauge sum. The third product, $R(Z_H,K_{DP})$, was showing the best performance of all the three compared and it was correlating well with gauge accumulation time series and one-month accumulation of 69.5 mm was just 0.6 mm lower than rain gauge sum.

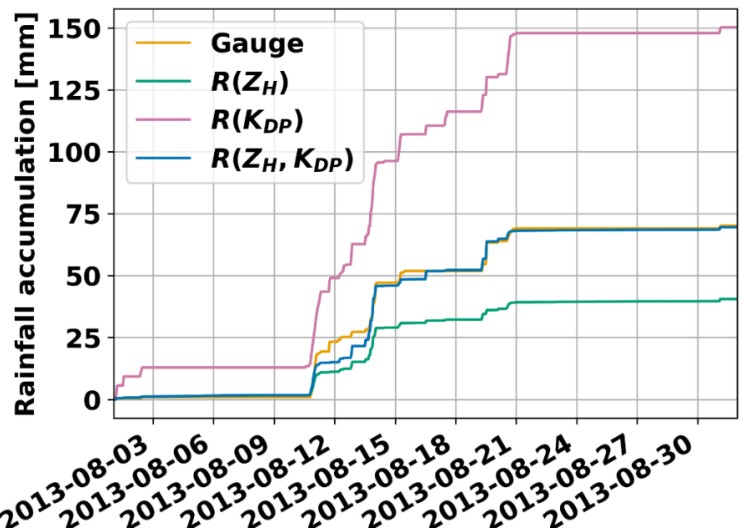

**Figure 2.** One month 1-hour rainfall cumulative accumulations, Sürgavere radar data, Jõgeva station gauge data. $Z_H$ not corrected for attenuation.

Gauge and radar accumulations are not always so well correlated as Fig. 3 demonstrates. In this accumulation period, there are rainfall events which show that gauge values can be both under- and overestimated by radar products. Rainfall around 11[th] of June 2016 is overestimated by all radar QPE products with the smallest overestimation by $R(Z_H)$ and greatest by $R(K_{DP})$ which overestimated the gauge by more than double in this event. In the following days until 21[st] of June 2016 light to medium precipitation was recorded by the gauge and during this time $R(K_{DP})$ mostly overestimated the gauge accumulations while $R(Z_H)$ underestimated rainfall. On the 21[st] of June 2016, a convective rainfall event occurred during which 51 mm of rainfall was measured in 2 hours with a gauge. All radar QPE products underestimated the rainfall amount during this event. By the end of the month-long accumulation period $R(Z_H,K_{DP})$ was closest to the gauge value (underestimation by 16.6 mm) while $R(Z_H)$ underestimated even more and $R(K_{DP})$ again overestimated gauge measurements.

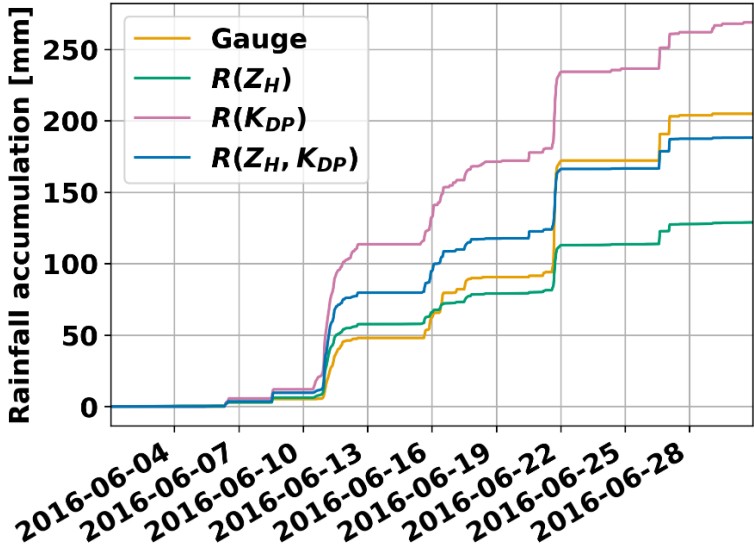

**Figure 3.** One month 1-hour rainfall cumulative accumulations, Sürgavere radar data, Tartu-Tõravere station gauge data. $Z_H$ not corrected for attenuation.

Figure 4 illustrates a case from Italy, a comparison of a gauge located within a 30 km distance from the radar to Bric della Croce radar precipitation estimation products. At the end of the 34-hour period, the specific differential phase based product $R(K_{DP})$ has the smallest error compared to gauge as it overestimates the gauge measurement of 40.6 mm by 2.0 mm. On the other hand, in light rain $R(K_{DP})$ is overestimating significantly - in the first 13 hours when a gauge measured 3.4 mm of accumulated rainfall it already estimated 12.2 mm. $R(Z_H)$ was underestimating even in light rain and in heavy rain, the difference compared to gauge measurement increased further. At the end of the period the underestimation was nearly threefold (15.6 mm compared to gauge accumulation of 40.6 mm). $R(Z_H,K_{DP})$ product showed good correlation with a gauge in light precipitation as it was mostly based on reflectivity data, but in the case of more intense precipitation, it was still underestimating compared to gauge data. At the end of the period, the accumulated value for $R(Z_H,K_{DP})$ was 26.7 mm.

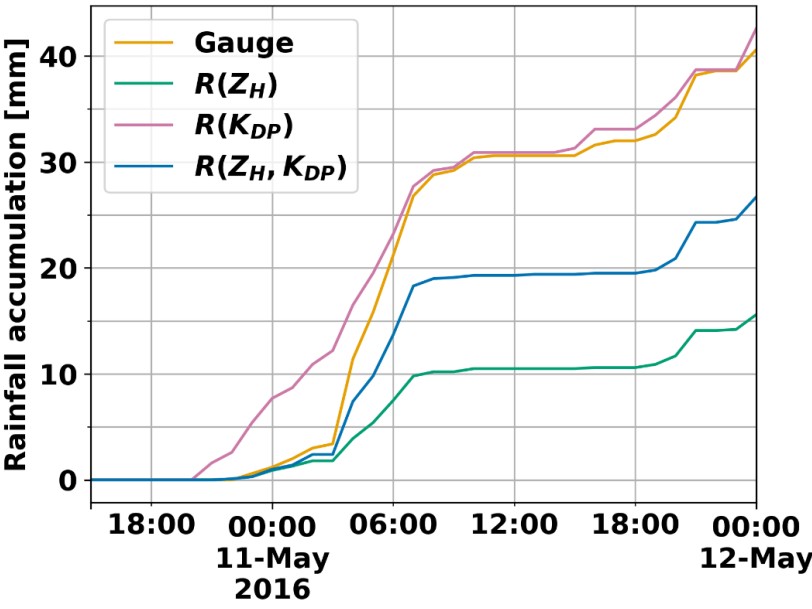

**Figure 4.** 1-hour rainfall cumulative accumulations from Verolengo gauge, located at 29 km from the radar, and co-located Bric della Croce radar QPE. $Z_H$ not corrected for attenuation.

In all selected cases the general behaviour of QPEs is similar. Weather radar estimations, even when sampled by 15-minutes interval observations, follow gauge measurements with good agreement. Although the second case from Estonia illustrated well that longer scan interval increases the scatter and particularly with small scale convective precipitation for which minimal sampling interval is the most beneficial. From Italy, the example case was much shorter, but the precipitation intensity was higher. In both cases, $R(K_{DP})$ generally overestimates precipitation amounts, especially in light rain cases. In Italy, the $R(K_{DP})$

overestimation is smaller. One of the causes of this behaviour might be more intense precipitation in Italy where $K_{DP}$ measurement became more accurate. More intense rainfall on the other hand caused greater underestimation of $R(Z_H)$ based precipitation accumulation from gauge values compared to Estonia. Another cause of differences between the two countries might be differences in the drop size distribution climatologies. Rainfall retrieval relations also entail errors and to keep the comparison as uniform as possible we decided to use the same relations for both Italy and Estonia. These example cases

demonstrated that radar can be used for 1-hour accumulations, but systematic errors cannot be excluded. These cases also presented the shortcomings of studies based only on a few cases. The performance of a QPE method depends heavily on a chosen case and it might perform differently on a long-term analysis. To find out errors and uncertainties and to see how QPEs compare to gauge measurements on a longer scale will be looked at in the next sections.

### 3.2 Comparison of one-hour accumulations

The quality of the rainfall estimates is compared at various accumulation intervals. Comparing different intervals can also be useful to point out representativeness issues caused by low radar scan rates. The investigated period covers the years 2011-2018 in Estonia and 2012-2016 in Italy.

First, in this section hourly accumulations are analysed. Hourly accumulations are especially important for small basins and in extreme precipitation climatology analysis. Hourly rainfall maxima can provide valuable data for flash flood nowcasting and

other hydrological applications.

**Table 3**. Verification of the radar-based rainfall 1-hour accumulation products of Estonia. $Z_H$ not corrected for attenuation.

|  | $R(Z_H)$ | $R(K_{DP})$ | $R(Z_H,K_{DP})$ |
|---|---|---|---|
| CC | 0.679 | 0.674 | 0.697 |
| NMAE | 0.537 | 0.868 | 0.594 |
| NMB | -0.143 | 1.861 | 0.298 |
| RMSE(mm) | 1.615 | 2.131 | 1.677 |
| NASH | 0.214 | -0.037 | 0.184 |

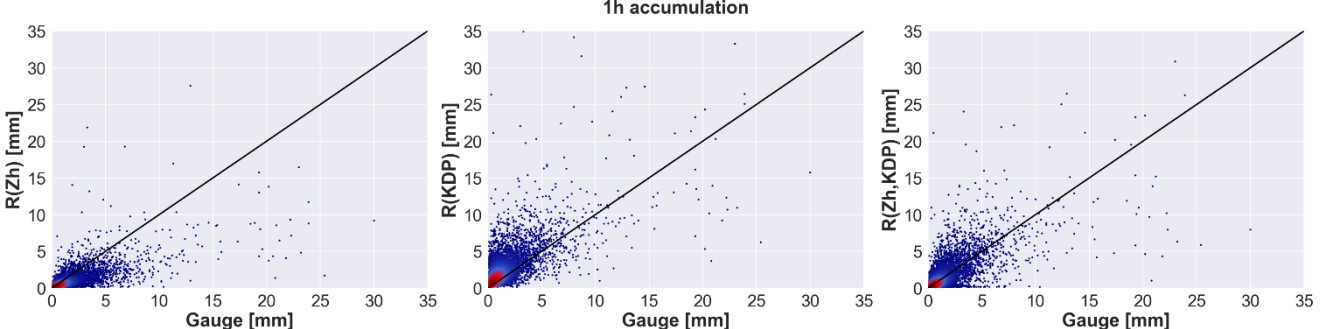

**Figure 5.** Scatter plots of radar-based rainfall estimates against rain gauge observations for 1-hour accumulation intervals in Estonia 2011-2018. The corresponding verification measures are presented in Table 3. The number of radar-gauge data pairs with 8 gauges and accumulations > 0.1 mm is 7,019. $Z_H$ not corrected for attenuation.

Table 3 presents the verification results for the hourly accumulation interval in Estonia. Figure 5 shows the corresponding scatter plots. As can be seen, the $R(Z_H)$ estimation generally underestimates rainfall, especially heavy events while it has the best error verification values (Nash-Sutcliffe Efficiency 0.214, NMAE 0.537, NMB -0.143, and RMSE 1.615 mm). $R(K_{DP})$ on the other hand overestimates accumulations for low-intensity events as could be presumed. $R(Z_H,K_{DP})$ shows considerable improvement by combining strong aspects of the two methods. It has the highest correlation coefficient (0.697) of all the products.

Nevertheless, it can be seen from the scatterplots that there is a lot of scatter in the hourly radar accumulations with all products. Mostly, it can be linked to the low spatial representativeness of the point measurements of rain gauges. This effect is more pronounced on a short time scale and it originates from a scarce gauge network and insufficient radar scan rate. Small scale effects like wind drift might also be more influential on a shorter accumulation period (Lauri et al., 2012). The reason why $R(Z_H)$ might have the best performances when NMAE and RMSE are considered is that there are not very many heavy rainfall cases in Estonia, and this tends to favour $R(Z_H)$ in the verification comparisons.

From Italian hourly accumulation scatterplots in Fig. 6, it can be seen that the overall behaviour of the radar products is similar to Estonia. Although from Fig. 6 it can be noticed that of the four highest 1-hour accumulations measured by the gauge, three of them have significantly higher radar estimates for $R(Z_H,K_{DP})$ than either $R(Z_H)$ or $R(K_{DP})$. This could be explained by precipitation that was very variable in intensity and also in spatial coverage in these three cases which in turn caused unsteady behaviour of the precipitation estimates. $Z_H$ underestimates high intensities, but with low intensities $K_{DP}$ becomes noisy and the rainfall intensity estimation is not feasible. Finally, to reduce $K_{DP}$ uncertainties range averaging is mandatory, leading to underestimation in case of very localized showers. By blending both $R(Z_H)$ and $R(K_{DP})$, a better rainfall estimation is expected. Table 4 presents the corresponding verification results. $R(Z_H)$ underestimates, particularly at intense precipitation events. $R(K_{DP})$ generally overestimates hourly accumulations especially at low-intensity cases: as stated by Wang et al. (2013), $R(K_{DP})$ generates noisier estimations at low rain rates. $R(Z_H,K_{DP})$ outperforms both other products in Italy which is confirmed by verification metrics as it overcomes the shortcomings of the other estimations.

Less random scatter is visible in Italian hourly data due to the more frequent scan strategy. $R(Z_H)$ is underestimating more than in Estonia as expected because in Italy intense rainfall is more frequent - it has larger RMSE and even more negative NMB. Probably for the same reason $R(K_{DP})$ is more accurate in Italy than in Estonia as it has smaller NMAE and NMB while having larger RMSE due to higher rainfall intensities recorded in Italy.

**Table 4**. Verification of the radar-based rainfall 1-hour accumulation products of Italy. $Z_H$ not corrected for attenuation.

|          | $R(Z_H)$ | $R(K_{DP})$ | $R(Z_H,K_{DP})$ |
|----------|----------|-------------|-----------------|
| CC       | 0.843    | 0.808       | 0.870           |
| NMAE     | 0.531    | 0.514       | 0.423           |
| NMB      | -0.296   | 0.678       | 0.120           |
| RMSE(mm) | 3.136    | 3.037       | 2.750           |
| NASH     | 0.364    | 0.385       | 0.443           |

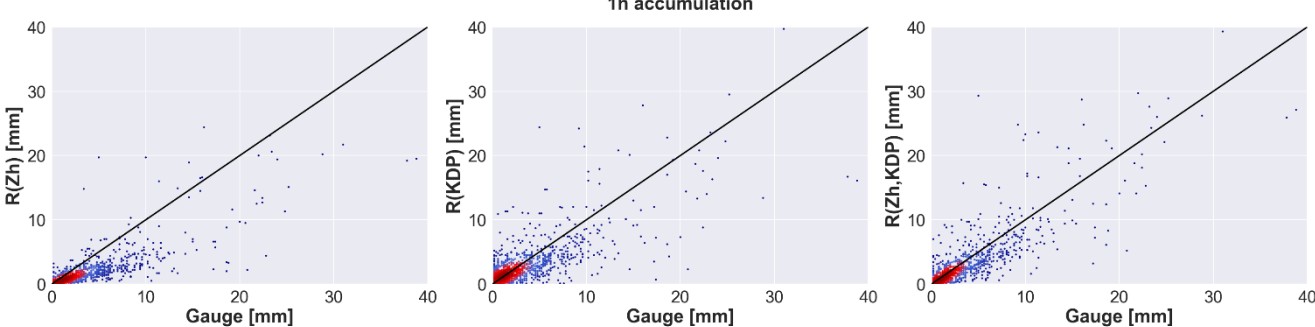

**Figure 6.** Italy 1-hour accumulations 2012-2016. The corresponding verification measures are presented in Table 4. The number of radar-gauge data pairs with 42 gauges and accumulations > 0.1 mm is 1,233. $Z_H$ not corrected for attenuation.

### 3.3 Comparison of 24-hours accumulations

Table 5 shows the verification results for the daily accumulation interval in Estonia, while Fig. 7 presents the corresponding scatter plots. As expected, much less scatter can be seen than on the daily level but overall, the results are consistent with the hourly interval verification outcomes. Using longer accumulation intervals leads to less severe errors as the longer period compensates for both underestimates and overestimates. Reflectivity based product, $R(Z_H)$, is still underestimating rain depths while the negative bias is considerably smaller than in hourly interval data. By looking at the definition of NMB in Eq. (3) it

can be seen that in case the same underlying samples are used NMB should be equal on all accumulation lengths. In our study, the underlying samples were different as the 0.1 mm threshold was applied after the accumulation as the last step before calculating the verification metrics. This emphasizes the importance of low-intensity precipitation for total accumulations. $R(K_{DP})$ is the least accurate of the three products also on daily accumulation level with the lowest correlation and highest error scores. The combined product, $R(Z_H,K_{DP})$, removes the negative bias of $R(Z_H)$ and shows better correlation and substantial

improvement in terms of both the systematic error and the overall error compared to $R(K_{DP})$. $R(Z_H,K_{DP})$ has the smallest NMAE of 0.438, RMSE of 3.992 mm, and the highest Nash-Sutcliffe Efficiency equal to 0.392. Overall there is noticeably less scatter in the daily radar accumulations compared to the 1-hour interval.

**Table 5**. Verification of the radar-based rainfall 24-hours accumulation products of Estonia. $Z_H$ not corrected for attenuation.

|          | $R(Z_H)$ | $R(K_{DP})$ | $R(Z_H,K_{DP})$ |
|----------|----------|-------------|-----------------|
| CC       | 0.831    | 0.792       | 0.827           |
| NMAE     | 0.475    | 0.845       | 0.438           |
| NMB      | -0.050   | 2.290       | 0.343           |
| RMSE(mm) | 4.366    | 7.195       | 3.992           |
| NASH     | 0.335    | -0.097      | 0.392           |

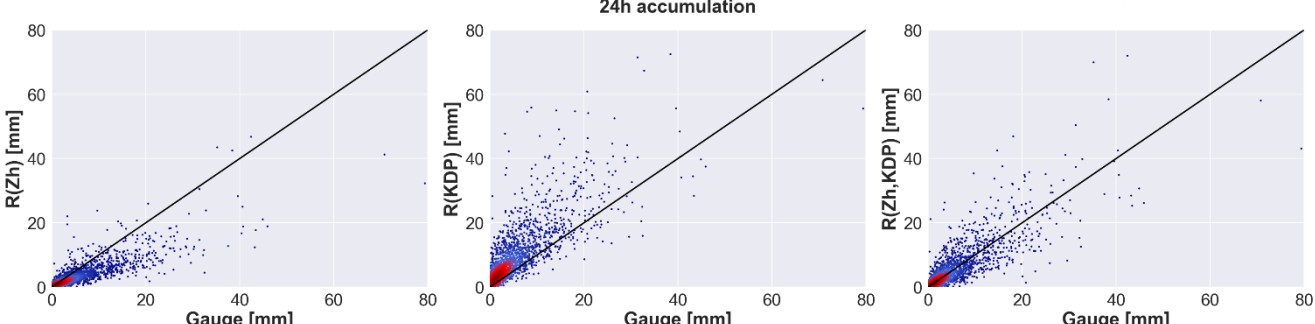

**Figure 7.** Estonia 24-hours accumulations 2011-2018. The corresponding verification measures are presented in Table 5. The number of radar-gauge data pairs with 8 gauges and accumulations > 0.1 mm is 2,148. $Z_H$ not corrected for attenuation.

Table 6 shows the verification results for the daily accumulation interval in Italy, while Fig. 8 presents the corresponding scatter plots. $R(Z_H)$ is slightly underestimating compared to gauge results and surprisingly it outperforms other competing products in all metrics except Pearson's correlation coefficient. $R(K_{DP})$ is again overestimating the most and has the lowest correlation with gauge data. $R(Z_H,K_{DP})$ notably improves the $R(K_{DP})$ on all verification metrics but does not exceed $R(Z_H)$

except for correlation coefficient which is the highest of all three products with r of 0.708. In Italy, the decrease in scatter of radar accumulations cannot be observed compared to the 1-hour level. In Fig 7. two regimes can be observed, and we assume that VPR correction leads to these regimes. Bric della Croce weather radar is located on a top of a hill at 770 m a.s.l. and during the winter season, a vertical profile reflectivity correction (VPR) is applied (Koistinen, 1991). This correction is manually switched on at the beginning of the cold season and it is switched off at the end. In the case of convective precipitation, this

correction may lead to rainfall overestimation. On the other hand, stratiform cold precipitation is heavily underestimated when VPR correction is switched off.

**Table 6**. Verification of the radar-based rainfall 24-hours accumulation products of Italy. $Z_H$ not corrected for attenuation.

|  | $R(Z_H)$ | $R(K_{DP})$ | $R(Z_H,K_{DP})$ |
| --- | --- | --- | --- |
| CC | 0.692 | 0.661 | 0.708 |
| NMAE | 0.504 | 0.636 | 0.553 |
| NMB | -0.01 | 0.789 | 0.459 |
| RMSE(mm) | 8.909 | 11.071 | 10.552 |
| NASH | 0.238 | 0.054 | 0.098 |

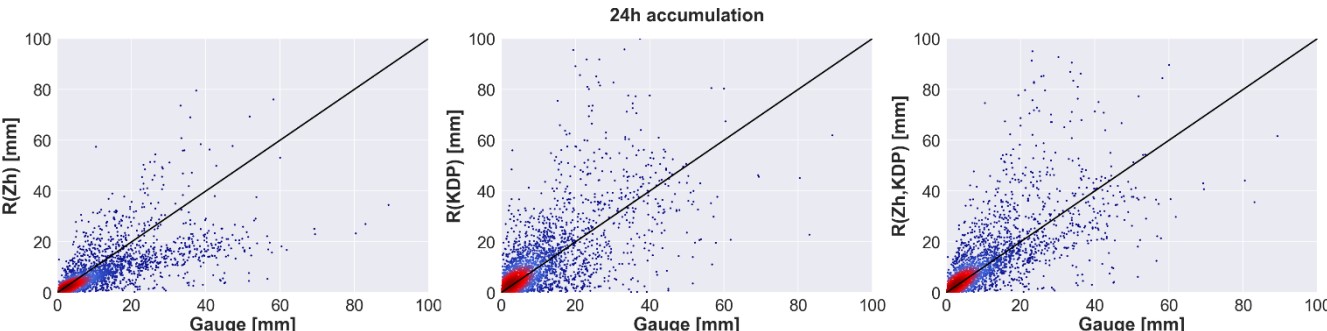

**Figure 8.** Italy 24-hours accumulations 2012-2016. The corresponding verification measures are presented in Table 6. The number of radar-gauge data pairs with 42 gauges and accumulations > 0.1 mm is 3,010. $Z_H$ not corrected for attenuation.

### 3.4 Comparison of monthly accumulations

Table 7 shows the verification results for the monthly accumulation interval in Estonia, while Fig. 9 presents the corresponding scatter plots. Compared to shorter time scales overall on a monthly scale the correlation of all the products with gauge accumulations is higher. $R(Z_H)$ is underestimating with a larger mean bias (-0.284) than on a daily level but with a smaller normalized mean absolute error (0.360). $R(K_{DP})$ is showing less scatter than on shorter time scales like other products while still heavily overestimating accumulations (NMB equal to 1.042 with RMSE equal to 62.466 mm). On the monthly

accumulation level $R(Z_H, K_{DP})$ outperforms the two other products to a great extent. It is well correlated to gauge values with small scatter as it is performing great both in low and high accumulation cases. The correlation coefficient is nearly identical to $R(Z_H)$, but it removes the systematic underestimation of $R(Z_H)$ and overestimation of $R(K_{DP})$ and exceeds them in all other verification metrics.

**Table 7**. Verification of the radar-based rainfall monthly accumulation products of Estonia. $Z_H$ not corrected for attenuation.

|          | $R(Z_H)$ | $R(K_{DP})$ | $R(Z_H,K_{DP})$ |
|----------|----------|-------------|-----------------|
| CC       | 0.877    | 0.789       | 0.875           |
| NMAE     | 0.360    | 0.822       | 0.214           |
| NMB      | -0.284   | 1.042       | 0.109           |
| RMSE(mm) | 27.448   | 62.466      | 16.704          |
| NASH     | 0.155    | -0.924      | 0.486           |

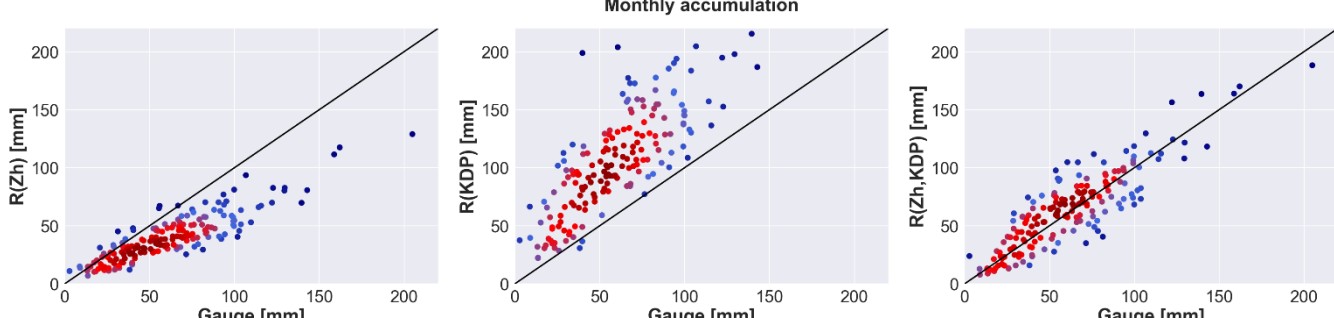

**Figure 9.** Estonia monthly accumulations 2011-2018. The corresponding verification measures are presented in Table 7. The number of radar-gauge data pairs with 8 gauges is 179. $Z_H$ not corrected for attenuation.

Table 8 shows the verification results for the monthly accumulation interval in Italy, while Fig. 10 presents the corresponding scatter plots. Scatterplots reveal similar characteristics to the daily level accumulations of the products. $R(Z_H)$ is

420 underestimating rainfall also on a monthly scale and $R(K_{DP})$ overestimating. $R(Z_H, K_{DP})$ is still overestimating but with a decreased RMSE compared to $R(K_{DP})$ product. It also exhibits the highest correlation coefficient of the three. According to the verification results, most of the metrics indicate better performance of the radar products on a monthly scale compared to daily intervals. The correlation coefficient is higher and NMAE is lower on all the products when the two timescales are compared.

**Table 8**. Verification of the radar-based rainfall monthly accumulation products of Italy. $Z_H$ not corrected for attenuation.

|          | $R(Z_H)$ | $R(K_{DP})$ | $R(Z_H,K_{DP})$ |
|----------|----------|-------------|-----------------|

| | | | |
|---|---|---|---|
| CC | 0.776 | 0.726 | 0.799 |
| NMAE | 0.375 | 0.488 | 0.408 |
| NMB | -0.128 | 0.310 | 0.337 |
| RMSE(mm) | 23.737 | 30.802 | 24.914 |
| NASH | 0.288 | 0.076 | 0.253 |

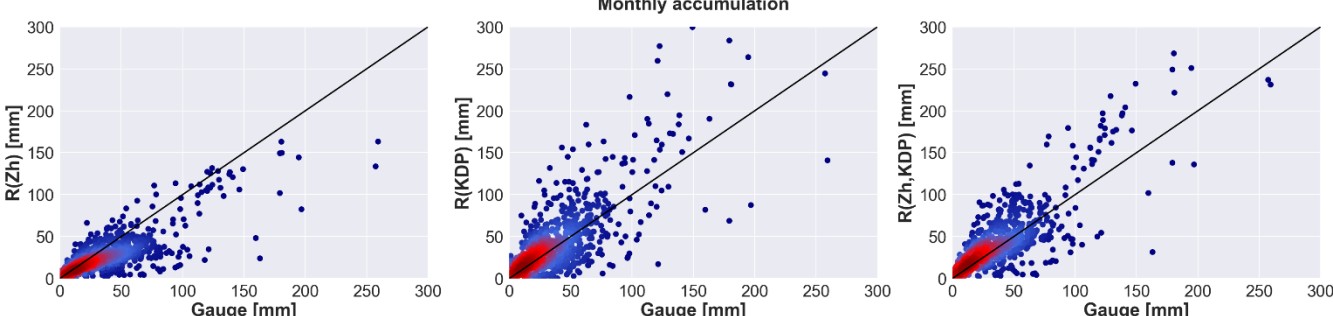

**Figure 10.** Italy monthly accumulations 2012-2016. The corresponding verification measures are presented in Table 8. The number of radar-gauge data pairs with 42 gauges is 675. $Z_H$ not corrected for attenuation.

## 4. Conclusions

In the present study polarimetric rainfall retrieval methods for the fully operational C-Band radars in Sürgavere, Estonia, and Bric della Croce, Italy have been analysed. The study focuses on the warm period of the year and a long period of multi-year data is used. From Estonia five years of data from 2011 to 2018 has been included, from Italy, the data interval ranges from 2012 to 2016. Reflectivity data were calibrated following a self-consistency theory and measured horizontal reflectivity ($Z_H$) was corrected accordingly. To calculate rainfall from polarimetric variables, the differential propagation phase ($\phi_{DP}$) was reconstructed and based on that specific differential phase ($K_{DP}$) retrieved. To achieve this the transparently implemented algorithm phase_proc_lp (Giangrande et al., 2013) in the open-source toolkit Py-ART was used for Estonian data. For Italian data, $K_{DP}$ precipitation estimates were obtained following the theory set down in Wang et al. (2009).

Three radar rainfall estimation products were computed: horizontal reflectivity based product $R(Z_H)$, specific differential phase based product $R(K_{DP})$, and a combined product based on the previous two $R(Z_H,K_{DP})$. Rain gauge network data of Italy and Estonia were used as ground truth. 1-hour, 24-hours, and monthly accumulations were derived from the radar products and gauge data.

Time-series comparison revealed that even with 15-minute scan interval radar is suitable for QPE, at least with more widespread precipitation like stratiform rain. Still, on the shortest accumulation period of 1-hour, the scarcer radar data from Estonia had more scatter than data from Italy where the scan interval was 10 minutes on older data and 5 minutes since 2013. As an overall trend, the longer the accumulation period the less scattering was visible. The case comparisons revealed also the shortcomings of analysis based only on selected short periods. The performance of the QPE methods then depends on the representativeness of the chosen cases and results can easily be skewed. Using a dataset with a length of at least several years without preselection provides more robust results and allows for evaluating the operational usability of the methods.

When the three products are compared to each other based on the full length of 5 years of data in the case of Estonia the $R(Z_H,K_{DP})$ was superior to $R(Z_H)$ and $R(K_{DP})$ on all accumulation periods. Especially on the monthly accumulation scale it was

performing distinctly better as it had RMSE 39% lower than the nearest competitor, the $R(Z_H)$ product, and even 73% lower than $R(K_{DP})$. In Italy, the $R(Z_H,K_{DP})$ product was exceeding the two others clearly on an hourly level. On 24-hours and monthly accumulation scale, it had the highest correlation with gauge measurements but the error verification measures were slightly higher than those of the $R(Z_H)$. Nevertheless, it outperformed $R(K_{DP})$ on all timescales.

Overall the results show that the combined product $R(Z_H,K_{DP})$ performs better on almost all of the verification measures in both countries compared to $R(Z_H)$ and $R(K_{DP})$ as it uses successfully the benefits of each other product and eliminates the weaknesses. $R(Z_H)$ was good at low precipitation intensities but in general, it was underestimating precipitation. It had an average NMB of -0.159 for all accumulation lengths in Estonia and -0.145 in Italy. $R(K_{DP})$ was performing well at higher intensities but in general was overestimating precipitation. It had an average NMB of 1.731 for all the accumulation lengths in Estonia and 0.592 in Italy. While the combined product $R(Z_H,K_{DP})$ was slightly overestimating precipitation with an average NMB of 0.250 for all the accumulation lengths in Estonia and 0.305 in Italy. In both countries the $R(Z_H,K_{DP})$ product also had the highest average CC over all the accumulation lengths with CC of 0.800 in Estonia and 0.792 in Italy. Generally, the CC was higher the longer the accumulation period was with the highest CC in monthly accumulations ($R(Z_H,K_{DP})$ CC of 0.875 in Estonia and 0.799 in Italy).

In Estonia, the overestimation of $R(K_{DP})$ was noticeably higher than in Italy. We hypothesize that this is mostly due to different climatological regimes between Italy and Estonia as high-intensity rainfalls occur more frequently in Italy. Although one has to keep in mind that the radars were from different manufacturers and thus also the used $K_{DP}$ retrieval algorithms were different which might be the cause of some discrepancy. Another source of error might originate from the implemented $Z_H$-$R$ and $K_{DP}$-$R$ relations which might not perform equally in different climates. Overall the results of the study showed that dual polarimetric radar QPE and especially the combined product $R(Z_H,K_{DP})$ show good potential to be used on long-term datasets if certain limitations are considered.

Synoptic patterns could be used as an additional source for classifying the radar accumulations. This would enable to verify the performance of each radar product on stratiform and convective events. Moreover, it could be used to investigate if frequent scans play a bigger role in convective events than stratiform as could be hypothesized and to quantify the effect.

For future studies, it would also be useful to calculate probabilities and return periods of extreme rainfall for weather radar-based rainfall climatology.

*Code and data availability.* The code used to conduct all analyses in this paper is available by contacting the authors. Gauge and radar data used in this study are available by contacting the authors.

*Author contributions.* TV, RC, PP, and DM directly contributed to the conception and design of the work. TV and RC collected and processed the various datasets and wrote the original draft with input from PP and DM. All authors reviewed and edited the final draft.

*Competing interests.* The authors declare that they have no conflict of interest.

*Acknowledgements.* This work was partly supported by the project IUT20-11 of the Estonian Ministry of Education and Research, the Estonian Research Council grant PSG202, and the European Regional Development Fund within National Programme for Addressing Socio-Economic Challenges through R&D (RITA1/02-52-07).

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
