# Peer review of "Evaluation of the dual-polarization weather radar quantitative precipitation estimation using long-term datasets"

_Hydrology and Earth System Sciences, 2019_

## Referee Comment (RC1) · Anonymous Referee #1 · 6 Feb 2020

Review of Manuscript hess-2019-624 Use of dual-polarization weather radar quantitative precipitation estimation for climatology Voormansik et al., 2020

GENERAL COMMENTS

This study presents an evaluation of quantitative precipitation estimates based on dual-polarization radar measurements for 1h, 24h, and one-month durations. It is based on relatively long radar datasets collected from two radars located in two different places with different climate conditions. The results show the added value of dual-pol rainfall estimates compared with the traditional method based on the horizontal reflectivity only.

The focus on the paper is clearly on the evaluation of the performance of the method and as mentioned in the abstract the main application is hydrological forecast and early

warning system. The use for climatology is not addressed and the datasets are actually not long enough to derive climatological information. I would recommend to change the title of the paper to reflect the actual scope of the study.

The paper is well organized and the study is relevant for the scientific community. However, there are some weaknesses and, in my view, the paper requires a major revision before publication. I recommend the following improvements:

- The description of the state of the art should be extended. Very little reference is made to previous studies on the evaluation of QPE based on dual-polarization measurements

- The description of the radar processing must be improved. Very little is said on the choice of various settings and parameters. Some tuning has been applied but without explained how it has been performed.

- The impact of some settings in the selection of the dataset and in the method for comparing and evaluating the various QPE methods should be tested.

- I would recommend to test the use of horizontal reflectivity without re-calibration based on dual-polarization data. This would allow to point out the benefit of such re-calibration.

- The impact of the 5-min to 15-min temporal sampling is addressed but the present study does not allow to isolate this effect from many other factors influencing the quality of the QPE. In the specific comments hereafter I propose a simple method that would allow to evaluate this impact. I recommend to test it.

- The main results of the study should be better presented in the abstract and the conclusion. What are the most original results of the study ?

SPECIFIC COMMENTS

Abstract

The length of the datasets should be mentioned in the abstract.

The use for climatology is not mentioned in the abstract and it is indeed not the main focus of the study.

The abstract should shortly present the main results of the study.

1. Introduction

Satellite-based rainfall estimates are not only limited by the resolution but also by the accuracy of the estimates.

The dataset starts in 2011. This record is probably long enough to perform an evaluation of the quality but still too short to derive robust climatological information. Climatology is certainly one of the future applications of radar-based QPEs (e.g., Saltikoff et al., BAMS, 2019) and it should be mentioned here as one of the applications of QPEs next to nowcasting, hydrological forecasts or agriculture. Thera are very few references to similar studies evaluating the quality of dual-pol based QPE.

2. Data and methods

2.1 Rain gauge measurements

Can you shortly describe how the measurements are quality-controlled?

L74 : why and how is this subset selected ?

2.2 Weather radar precipitation estimates

One of the benefits of dual-pol measurements is the reduction of ground clutter. Is there any clutter filtering based on these measurements in the processing ?

L85 : why are KDP measurements unreliable at short range ?

L100: what happens after 2016 ?

The processing of the raw PHIDP data to derive KDP is only very briefly described. Some parameters have been tuned but we don't know which and how. Wat is the impact of this tuning on the final results ? Is there any impact of the PHIDP processing

on the resolution in range ? Is the final resolution appropriate for estimating heavy rainfall from convective cells with relatively small spatial extent ? More must be said on how the optimal settings have been determined. Is the dataset used for verification independent of the dataset used for tuning ?

The re-calibration of the horizontal reflectivity using the self-consistency theory should be a bit more explained even if a detailed description is available elsewhere. For example, is there also some fine-tuning in this re-calibration ? The re-calibration is another benefit of dual-pol measurements and it would be interesting to show what is the impact on the quality of the derived QPEs. Comparisons of QPE derived from horizontal reflectivity with and without re-calibration would be very interesting. I recommend to include these comparisons.

L 127 : how is the 25 dBZ threshold selected ?

2.3 Comparison framework

L 137 : 30 km seems very small. Why such a limited study area ?

L 139 : hail is not considered as as possible precipitation type. Is this valid for Estonia ? In the description of the comparison framework, nothing is said about the minimum rainfall amounts used for the selection of the valid pairs and the production of the statistics. A threshold of 0.1 mm is mentioned in the legend of the figures. Is this threshold used all through the study ? It seems very small which means that some statistics might be strongly influenced by very small rainfall amounts. How do you apply this threshold ? Should gauge and QPE values both exceed 0.1 mm to make the pair valid ?

3. Results and discussion

3.1 Case comparisons

L157- 159 : unclear formulation

Figure 2 : the agreement between gauge and R(ZH,KDP) is almost perfect for this particular month. Does it give a realistic view on the results obtained in Estonia ? Perhaps showing a few additional cases (perhaps, as a supplement) would allow to get a better picture of the overall agreement between gauge and QPE values ?

L188 – 196 : Can you further elaborate on random versus systematic errors . As statement like "Systematic errors cannot be excluded" seems somewhat obvious when it concerns radar-based rainfall estimates. In the paper, the word "randomness" seems to be used for expressing "scatter".

L220. Many factors influence the scatter. The temporal sampling is one of them and the results shown here do not allow to isolate this effect. A proper way to test the impact of the temporal sampling on the scatter is possible with the Italian radar which produces a 5-min sampling dataset. A degraded dataset with 15-min temporal sampling can be produced by removing 2 out of 3 date files. The results obtained using the original 5-min and the degraded 15-min dataset would allow evaluating the impact of the temporal sampling.

Figure 7 : two regimes seem to appear. Can you comment on this ?

Figure 8 :why is a contour plot used here and not in the other figures ?

Conclusion

L 306 : A fourth radar rainfall estimate would be useful : R(ZH) without re-calibration based on dual-pol data.

L 327-329 : the formulation is not very clear. What do you mean with "filtering the radar accumulations" ? It seems also that the conclusion is known before performing the study.

The conclusion does not make clear what are the original results of the present study.

TYPOS AND FORMULATIONS

Strange formulations and spelling errors are present throughout the text. Some are listed below. I would recommend having the text proofread by a native English speaker.

L 12 and further : precipitation without s all through the text

L16 : legacy ?

L 30 : to a good effect ?

L97 : central respect Piemonte : strange formulation

---

## Referee Comment (RC2) · Hidde Leijnse (Referee) · 12 Feb 2020

In this paper long-term datasets from two different regions (Estonia and Northern Italy) are used to evaluate the performance of polarimetric weather radar quantitative precipitation estimates. Several years of radar and gauge data are used for this. This is a very interesting topic that is highly relevant and timely as long-term high-quality operational polarimetric datasets are becoming more and more available. The paper has a clear focus, which makes it pleasant to read. Some of the English used in the paper could be improved, but it certainly does not prohibit full understanding of the paper. I do have some questions that I would like to see clarified and some suggestions for improvements. In particular, I think there may be an error in at least one of the figures that I think the authors should look at. I think the paper should be published after major

revisions. Specific comments are given below.

**Specific comments**

1. I very much appreciate the honesty of the authors about removing low-quality data from the dataset that resulted from radar issues. Because of this, and because of the focus on only the warm season, I doubt whether mentioning "climatology" in the title would be suitable. Please reconsider this, or at least add to the title that this is about warm-season precipitation (which is still very valuable).

2. On line 72, it is stated that the Italian rain gauges have a resolution of 0.2 mm and 1 minute. This means that, if the gauges report rainfall intensities, the minimum rainfall intensity that these gauges would be able to record is 12 mm h$^{-1}$. Or do the gauges record total rainfall accumulation with a 1-minute time resolution (in which case there is no issue with the total accumulations)?

3. On lines 105-107, the computation of $\Phi_{\mathrm{DP}}$ and $K_{\mathrm{DP}}$ from raw $\Phi_{\mathrm{DP}}$ is mentioned, along with the fact that "carefully tuned parameter values according to data specifics" are used for this. It would be interesting and highly relevant to include a more thorough description of this in the paper, especially since $K_{\mathrm{DP}}$ is a key variable in this paper. I think a one- or two-sentence summary of the method would be nice, along with a short description of the parameters and how they were determined.

4. On lines 107-110, the self-consistency method for re-calibrating $Z_{\mathrm{H}}$ is discussed, where $Z_{\mathrm{DR}}$ is also used. Later, on lines 124-125, it is mentioned that the use of $Z_{\mathrm{DR}}$ for quantitative precipitation estimation is not recommended for C-band radars. I think it should be discussed here why $Z_{\mathrm{DR}}$ can be used for re-calibration of $Z_{\mathrm{H}}$.

5. On line 127, the threshold for switching between $R(Z_H)$ and $R(K_{DP})$ is defined to be 25 dBZ for $Z_H$. Using Eqs (1) and (2), this threshold translates to $R \approx 1$ mm h$^{-1}$ and $K_{DP} \approx 0.015°$ km$^{-1}$. These values are much lower than what is cited from the literature ($R = 50$ mm h$^{-1}$ and $K_{DP} = 0.5 - 1°$ km$^{-1}$). What is the reason for using this much lower threshold? I think this should be explained in the paper.

6. In Section 2.2, there is no mention of attenuation that could affect the $R(Z_H)$ estimates. This attenuation could be corrected for using $\Phi_{DP}$. Is there a specific reason why attenuation correction is not carried out?

7. In Section 2.2, it would be good to mention that the effect of VPR will be limited in the analyses because only data from the warm season will be used, and that only data close to the radars (70 km and 30 km for Estonia and Italy, respectively) will be used.

8. In Section 2.2, it is not explicitly mentioned how precipitation accumulations are computed. I assume (also based on the rest of the paper) that they are computed by simply adding subsequent instantaneous radar QPE values, without any space-time interpolation. It would be good to mention that here explicitly.

9. Is my interpretation of Fig. 1 correct if I say that in Estonia only a circular area around the radar is used (up to 70 km range), while in Italy a rectangular area $(60 \times 60$ km$^2)$ around the radar is used? If this is correct, is there an explanation of why two different areas have been used? This should be included in the paper.

10. Equation (3) for Pearson's correlation coefficient is incorrect. It should be:

$$CC = \frac{\sum_{i=1}^{n}(r_i - r_m)(g_i - g_m)}{\sqrt{\sum_{i=1}^{n}(r_i - r_m)^2}\sqrt{\sum_{i=1}^{n}(g_i - g_m)^2}}.$$

11. For the definition of the normalized mean error in Eq. (4), the multiplication with 100% needs to be omitted in order to make it consistent with the results presented in Tables 1-6. I would also like to suggest renaming this statistic to the "normalized mean absolute error", which in my view is closer to what it actually is.

12. The authors could consider to also normalize the RMSE in Eq. (6) with the mean gauge rainfall. In this way, all statistics will be dimensionless. This is of course just a choice, and I would also be perfectly fine with leaving the definition as it is.

13. On lines 192-193, the cause for the more severe underestimation of $R$ from $Z_{\mathrm{H}}$ in Italy than in Estonia is said to be the fact that there is more intense precipitation. However, doesn't this mean that the employed $Z - R$ relation is not suitable? Differences in raindrop size distribution (DSD) climatologies between Estonia and Italy may also cause differences. So it would be good to comment here on the suitability of the retrieval relations (Eqs (1) and (2)) for both regions.

14. On lines 198-199, it is stated that using different time intervals can help in understanding the effect of temporal sampling differences between radar and gauges. While this is certainly true, it should also be noted that using longer accumulation intervals will also lead to less severe errors (compensating underestimates and overestimates; the $R(K_{\mathrm{DP}})$ curve in Fig. 3 is a good example of this). I think a remark about this should also be added to the text. The same holds for line 219.

15. On lines 215-217, an important statement is made about the improvement that $R(Z_H, K_{DP})$ gives over the other methods. At first reading, I thought that this statement is too bold given the results presented in Table 1, but on second thought it is correct. What would have helped me is if something along the lines of "(i.e., each statistic is approximately as good at the best of the other two)" after "...other product's weak points" would have been included. You could consider including this here.

16. In Fig. 5, it is interesting to see that of the 4 highest 1-hour accumulations measured by a gauge, 3 of them have significantly higher radar estimates for $R(Z_H, K_{DP})$ than either $R(Z_H)$ or $R(K_{DP})$. This means that for $R(Z_H, K_{DP})$, probably the best estimator of $R$ is selected for most of the intervals (i.e., for at least one of the underlying 5-minute intervals $R(Z_H)$ is higher than $R(K_{DP})$, and it is correctly selected for $R(Z_H, K_{DP})$). I think this merits some more discussion in the paper, especially since this is the case for 3 of the 4 highest 1-hour accumulations.

17. On lines 242-243, it is stated that the normalized bias is much smaller for the 24-hour accumulations than for the 1-hour accumulations. However, looking at the definition of the NMB in Eq. (5), there should be absolutely no difference between the two, if the same underlying samples have been used (i.e., it makes no difference whether you first sum over 24 hours, and then subtract gauge from radar sums, or if you compute the difference first and then sum over 24 hours because subtraction is a linear operation). So what is the cause of these differences? Is it because you use different underlying samples, possibly by taking only accumulations above 0.1 mm (see captions of Figs 4 to 7)? If this is the case, this stresses the importance of low-intensity rain for total rainfall accumulations. This should be explained clearly. The same holds for differences between 24-hour and 1-month accumulations.

18. If I compare the NMB presented for $R(Z_H)$ in Table 3 and the corresponding panel in Fig. 6, I'm surprized at the fact that the underestimation by the radar is so small. Is this because there is an extremely high density of points just above the black line close to 0 mm in Fig. 6?

19. Comparison of Figs 5 and 7 gives me the feeling that there may be an error in one of them. For example, if I roughly add all of the accumulations from $R(Z_H)$ in Fig. 5, the resulting amount of rain is much smaller than when I roughly add all of the accumulations from $R(Z_H)$ in Fig. 7. Furthermore, the number of accumulations exceeding 0.1 mm given in the caption is higher for Fig. 7 than for Fig. 5. This is impossible unless a different dataset has been used. So I suggest to take another careful look at the figures and the results presented in the tables.

20. Figure 7 seems to show two regimes for $R(Z_H)$, where one overestimates and the other underestimates for higher rainfall accumulations. It would be interesting to discuss this in the paper. I'm interested to learn if these regimes are separable by some other variable such as time, temperature, or something else.

---

## Author Comment (AC1) · 9 Apr 2020

**Responses to comments from Anonymus Referee #1**

On „Use of dual-polarization weather radar quantitative precipitation estimation for climatology" by Tanel Voormansik et al.(HESS-2019-624)

**Referee's comment**

GENERAL COMMENTS

This study presents an evaluation of quantitative precipitation estimates based on dualpolarization radar measurements for 1h, 24h, and one-month durations. It is based on relatively long radar datasets collected from two radars located in two different places with different climate conditions. The results show the added value of dual-pol rainfall estimates compared with the traditional method based on the horizontal reflectivity only.

The focus on the paper is clearly on the evaluation of the performance of the method and as mentioned in the abstract the main application is hydrological forecast and early warning system. The use for climatology is not addressed and the datasets are actually not long enough to derive climatological information. I would recommend to change the title of the paper to reflect the actual scope of the study.

The paper is well organized and the study is relevant for the scientific community. However, there are some weaknesses and, in my view, the paper requires a major revision before publication. I recommend the following improvements:

- The description of the state of the art should be extended. Very little reference is made to previous studies on the evaluation of QPE based on dual-polarization measurements

- The description of the radar processing must be improved. Very little is said on the choice of various settings and parameters. Some tuning has been applied but without explained how it has been performed.

- The impact of some settings in the selection of the dataset and in the method for comparing and evaluating the various QPE methods should be tested.

- I would recommend to test the use of horizontal reflectivity without re-calibration based on dual-polarization data. This would allow to point out the benefit of such re-calibration.

- The impact of the 5-min to 15-min temporal sampling is addressed but the present study does not allow to isolate this effect from many other factors influencing the quality of the QPE. In the specific comments hereafter I propose a simple method that would allow to evaluate this impact. I recommend to test it.

- The main results of the study should be better presented in the abstract and the conclusion. What are the most original results of the study ?

**Authors' response**

Authors would like to sincerely thank the referee for the time and effort spent in reading the initial manuscript and for making many clear and constructive suggestions for improvement. This helped a lot to improve the manuscript.

SPECIFIC COMMENTS

Abstract

**Referee's comment**

The length of the datasets should be mentioned in the abstract.

**Authors' response**

Agreed. Sentence about the length of the datasets added to the abstract.

**Referee's comment**

The use for climatology is not mentioned in the abstract and it is indeed not the main focus of the study.

The abstract should shortly present the main results of the study.

**Authors' response**

We agree with the comments. The short conclusion of main results was added to the abstract:

"Overall the radar products showed similar results in Estonia and Italy when compared to each other. The product where radar reflectivity and specific differential phase were combined based on a threshold exhibited the best agreement with gauge values on all accumulation periods. In both countries reflectivity based rainfall quantitative precipitation estimation underestimated and specific differential phase based product overestimated gauge measurements in general."

**Referee's comment**

1. Introduction

Satellite-based rainfall estimates are not only limited by the resolution but also by the accuracy of the estimates.

**Authors' response**

Agreed. Added short description with reference about the accuracy of the estimates to the manuscript:

"What is more, satellite-based precipitation estimates are limited by the accuracy of the estimates. The accuracy of the estimates has regional dependency and therefore can vary due to physiography of the study areas (e.g. precipitation climate, land use and geomorphology) (Petropoulos and Islam, 2017)."

**Referee's comment**

The dataset starts in 2011. This record is probably long enough to perform an evaluation of the quality but still too short to derive robust climatological information. Climatology is certainly one of the future applications of radar-based QPEs (e.g., Saltikoff et al., BAMS, 2019) and it should be mentioned here as one of the applications of QPEs next to nowcasting, hydrological forecasts or agriculture. Thera are very few references to similar studies evaluating the quality of dual-pol based QPE.

**Authors' response**

We agree that the dataset we had for the study is too short to derive robust rainfall climatology. Additional references to studies evaluating the quality of dual-pol based QPE were added:

"Previous studies where the benefits of dual polarimetric radar QPE have been shown are mostly based on selected short time periods or only single events (Wang and Chandrasekar, 2010; Chang et al., 2016; Cao et al., 2018)"

**Referee's comment**

2. Data and methods

2.1 Rain gauge measurements

Can you shortly describe how the measurements are quality-controlled?

**Authors' response**

Short description of the quality control process added to the manuscript.

**Referee's comment**

L74 : why and how is this subset selected ?

**Authors' response**

The rain gauge subset consists of gauges that are located within the range limit that is applied to the radar data which is explained in Section 2.3 where comparison framework is described.

**Referee's comment**

2.2 Weather radar precipitation estimates

One of the benefits of dual-pol measurements is the reduction of ground clutter. Is there any clutter filtering based on these measurements in the processing ?

**Authors' response**

Agreed. Short description of polarimetric filtering used on data added to the manuscript.

**Referee's comment**

L85 : why are KDP measurements unreliable at short range ?

**Authors' response**

To get reliable KDP estimations averaging among range bins is required. However, close to the antenna, stable and reliable observations are not available, due to both the antenna itself and TR-limiters response time (or the dual polar switch in case of alternate transmission). The explanation was added to the manuscript as well.

**Referee's comment**

L100: what happens after 2016 ?

**Authors' response**

Reworded the sentence so it would be unambiguously understood: "Bric della Croce observations used in the study range from 2012 to 2016 whereas observations from 2012 to 2013 are with ten-minutes interval and from 2013 to 2016 with five minutes interval time resolution."

**Referee's comment**

The processing of the raw PHIDP data to derive $K_{DP}$ is only very briefly described. Some parameters have been tuned but we don't know which and how. What is the impact of this tuning on the final results? Is there any impact of the PHIDP processing on the resolution in range? Is the final resolution appropriate for estimating heavy rainfall from convective cells with relatively small spatial extent? More must be said on how the optimal settings have been determined. Is the dataset used for verification independent of the dataset used for tuning?

**Authors' response**

Following the referee comment several sentences to describe the derivation of KDP were added to the manuscript:

"With default parameter values the rays where differential propagation phase folding occurred did not unfold correctly and thus the function did not produce correct specific differential phase values. In order to fix the folding issue function parameters self_const (self-consistency factor) and low_z (low limit for reflectivity – reflectivity below this value is set to this limit) had to be tuned. The default values were 60000.0 and 10.0 respectively and after testing with various combinations of various values the values 12000.0 and 0.0 were found to produce optimal results and therefore were chosen for final calculations."

**Referee's comment**

The re-calibration of the horizontal reflectivity using the self-consistency theory should be a bit more explained even if a detailed description is available elsewhere. For example, is there also some fine tuning in this re-calibration ? The re-calibration is another benefit of dual-pol measurements and it would be interesting to show what is the impact on the quality of the derived QPEs. Comparisons of QPE derived from horizontal reflectivity with and without re-calibration would be very interesting. I recommend to include these comparisons.

**Authors' response**

We agree that the paper would benefit from providing more details about the re-calibration method. As the comparisons of QPE with and without re-calibrated horizontal reflectivity would be out of the scope and focus of this paper we would not include it. Following the referee comment short explanation of the theory along with the used filtering thresholds was added to the manuscript:

"The method essentially compares the observed differential propagation phase ($\phi_{DP}^{obs}$) to a calculated theoretical differential propagation phase ($\phi_{DP}^{th}$). The data used for calibration had to be filtered using a number of restrictions: only data from June to September was allowed; data from 0.5° elevation and 10-70 km range only used; only bins where horizontal and vertical polarization channel correlation coefficient was over 0.92 were used; any bins where $\phi_{DP}$ was greater than 12° were removed; whole ray where reflectivity was greater than 50 dBZ was removed; whole ray where $Z_{DR}$ was greater than 3.5 dB was rejected; only rays where $\Delta\phi_{DP}^{obs}$ was greater than 8° and where the consecutive rain path was at least 10 km was used; any scans in which precipitation occurred on top of the radome were removed."

**Referee's comment**

L 127 : how is the 25 dBZ threshold selected?

**Authors' response**

The threshold was selected after testing on a few months dataset with various reflectivity levels and this provided the best correlation with gauges. Following the referee comment a short description was also added to the manuscript.

**Referee's comment**

2.3 Comparison framework

L 137 : 30 km seems very small. Why such a limited study area?

**Authors' response**

The applied range limit is aimed mainly at eliminating uncertainties due to complex orography, like shielding by the mountains. Up to 30 km from Bric della Croce terrain is relatively flat while beyond that mountains block most of the radar signal for lowest elevations. It is explained in manuscript Section 2.3.

**Referee's comment**

L 139 : hail is not considered as as possible precipitation type. Is this valid for Estonia? In the description of the comparison framework, nothing is said about the minimum rainfall amounts used for the selection of the valid pairs and the production of the statistics. A threshold of 0.1 mm is mentioned in the legend of the figures. Is this threshold used all through the study? It seems very small which means that some statistics might be strongly influenced by very small rainfall amounts. How do you apply this threshold? Should gauge and QPE values both exceed 0.1 mm to make the pair valid?

**Authors' response**

We agree that hail as solid precipitation type was overlooked. It is now added to the manuscript. A threshold of 0.1 mm is set and applied such that both gauge and radar QPE values must exceed this value to make the pair valid. It is used all through the study. This clarification is added to the manuscript.

**Referee's comment**

3. Results and discussion

3.1 Case comparisons

L157- 159 : unclear formulation

**Authors' response**

The formulation was changed so it would be more clearly understood.

**Referee's comment**

Figure 2 : the agreement between gauge and R($Z_H$,$K_{DP}$) is almost perfect for this particular month. Does it give a realistic view on the results obtained in Estonia? Perhaps showing a few additional cases (perhaps, as a supplement) would allow to get a better picture of the overall agreement between gauge and QPE values?

**Authors' response**

We agree that Figure 2 might leave unrealistic view of the results obtained in Estonia. Another case was added to the manuscript Section 3.1 where the agreement between radar QPE and gauge was not so perfect.

**Referee's comment**

L188 – 196 : Can you further elaborate on random versus systematic errors . As statement like "Systematic errors cannot be excluded" seems somewhat obvious when it concerns radar-based rainfall estimates. In the paper, the word "randomness" seems to be used for expressing "scatter".

**Authors' response**

Systematic errors can originate for example from radar hardware calibration or unsuitable Z-R relationship (the actual drop size distribution is different than assumed in the Z-R relationship). Random errors can originate for example from incomplete beam filling, high intensity small scale rainfall events not completely resolved by the radar (spatial and/or temporal) resolution. Following the referee comment the word "scatter" was used in the paper instead of "randomness".

**Referee's comment**

L220. Many factors influence the scatter. The temporal sampling is one of them and the results shown here do not allow to isolate this effect. A proper way to test the impact of the temporal sampling on the scatter is possible with the Italian radar which produces a 5-min sampling dataset. A degraded dataset with 15-min temporal sampling can be produced by removing 2 out of 3 date files. The results obtained using the original 5-min and the degraded 15-min dataset would allow evaluating the impact of the temporal sampling.

**Authors' response**

We agree with the explanation and description of the methodology provided by the referee but decided to not include it in the study because it would be out of the scope and main focus of this study. Long accumulation datasets comprised of many years even out the errors, even on shorter accumulation periods but especially on longer periods.

**Referee's comment**

Figure 7 : two regimes seem to appear. Can you comment on this ?

**Authors' response**

The reviewer is right. The Bric della Croce weather radar is located on a top of hill at 770 m asl and during the winter season a vertical profile reflectivity correction (VPR) is applied (Koistinen, 1991). This correction is manually switched on at the beginning of the cold season and it is switched off at the end. In case of convective precipitation, this correction may lead to rainfall overestimation. On

the other hand, stratiform cold precipitation is heavily underestimated when VPR correction is switched off. So, the VPR correction leads to these regimes. The separation between the two regimes could be obtained by reducing the study area even more, limiting the study to June, July and August. Unfortunately only the corrected reflectivity (including VPR) is available for studied years; later both corrected and uncorrected become available. The explanation was added to the manuscript as well.

**Referee's comment**

Figure 8 :why is a contour plot used here and not in the other figures ?

**Authors' response**

The same plotting function was used for all scatterplots (Python seaborn data visualization library function *kdeplot* with scatter), but only on Figure 8 the number of data points was low and distribution coarse enough to make contours clearly visible. We agree that the plots do not look uniform enough and we are going to remake them.

**Referee's comment**

Conclusion

L 306 : A fourth radar rainfall estimate would be useful : $R(Z_H)$ without re-calibration based on dual-pol data.

**Authors' response**

While we agree that it would allow direct comparison of the reflectivity based rainfall estimates we would still not include it in this study because it would not add enough value to the comparison of other radar QPE products. Also the comparison results and conclusions would depend very much on radar calibration quality and it was not the focus of this paper to evaluate this.

**Referee's comment**

L 327-329 : the formulation is not very clear. What do you mean with "filtering the radar accumulations" ? It seems also that the conclusion is known before performing the study.

**Authors' response**

Agreed. Reworded the sentences.

**Referee's comment**

The conclusion does not make clear what are the original results of the present study.

**Authors' response**

Agreed. The conclusion was improved to make main original results of the study stand out more clearly.

**Referee's comment**

TYPOS AND FORMULATIONS

Strange formulations and spelling errors are present throughout the text. Some are listed below. I would recommend having the text proofread by a native English speaker.

L 12 and further : precipitation without s all through the text

**Authors' response**

Agreed and corrected in manuscript.

**Referee's comment**

L16 : legacy ?

**Authors' response**

Agreed. Replaced the word "legacy" with a more suitable "'conventional".

**Referee's comment**

L 30 : to a good effect ?

**Authors' response**

The phrase was replaced with a word "successfully".

**Referee's comment**

L97 : central respect Piemonte : strange formulation

**Authors' response**

Agreed. Reworded the sentence to be more clear.

Cited references:

Cao, Q., Knight, M. and Qi, Y.: Dual-pol radar measurements of Hurricane Irma and comparison of radar QPE to rain gauge data, In Proceed. of the 2018 IEEE Radar Conference, Oklahoma City, OK, USA, 23-27 April 2018, 0496-0501, https://doi.org/10.1109/RADAR.2018.8378609, 2018

Chang, W.Y., Vivekanandan, J., Ikeda, K. and Lin, P.L.: Quantitative precipitation estimation of the epic 2013 Colorado flood event: Polarization radar-based variational scheme, J. Appl. Meteorol. Climatol., 55, 1477-1495, https://doi.org/10.1175/JAMC-D-15-0222.1, 2016

Koistinen, J.: Operational correction of radar rainfall errors due to the vertical reflectivity profile, in: Proceedings of the 25th Radar Meteorology Conference, American Meteorological Society, Paris, France, 91–96, 1991.

Petropoulos, G.P. and Islam, T.: Remote Sensing of Hydrometeorological Hazards. CRC Press, Boca Raton FL, USA, 2017.

Wang, Y. and Chandrasekar, V.: Quantitative precipitation estimation in the CASA X-band dual-polarization radar network, J. Atmos. Ocean. Technol., 27, 1665-1676, https://doi.org/10.1175/2010JTECHA1419.1, 2010.

---

## Author Comment (AC2) · 9 Apr 2020

**Responses to comments from Referee #2**

On „Use of dual-polarization weather radar quantitative precipitation estimation for climatology" by Tanel Voormansik et al.(HESS-2019-624)

**Referee's comment**

GENERAL COMMENTS

In this paper long-term datasets from two different regions (Estonia and Northern Italy) are used to evaluate the performance of polarimetric weather radar quantitative precipitation estimates. Several years of radar and gauge data are used for this. This is a very interesting topic that is highly relevant and timely as long-term high-quality operational polarimetric datasets are becoming more and more available. The paper has a clear focus, which makes it pleasant to read. Some of the English used in the paper could be improved, but it certainly does not prohibit full understanding of the paper. I do have some questions that I would like to see clarified and some suggestions for improvements. In particular, I think there may be an error in at least one of the figures that I think the authors should look at. I think the paper should be published after major revisions. Specific comments are given below.

**Authors' response**

Authors would like to sincerely thank the referee for the time and effort spent in reading the initial manuscript and for making many clear and constructive suggestions for improvement. This helped a lot to improve the manuscript.

**Referee's comment**

Specific comments

1. I very much appreciate the honesty of the authors about removing low-quality data from the dataset that resulted from radar issues. Because of this, and because of the focus on only the warm season, I doubt whether mentioning "climatology" in the title would be suitable. Please reconsider this, or at least add to the title that this is about warm-season precipitation (which is still very valuable).

**Authors' response**

We agree with the comment. We decided to change the title to be more appropriate considering the length of the dataset used. The new proposed title is "Applicability of dual-polarization weather radar quantitative rainfall estimation for climatological purposes".

**Referee's comment**

2. On line 72, it is stated that the Italian rain gauges have a resolution of 0.2 mm and 1 minute. This means that, if the gauges report rainfall intensities, the minimum rainfall intensity that these gauges would be able to record is 12 mm h$^{-1}$ . Or do the gauges record total rainfall accumulation with a 1-minute time resolution (in which case there is no issue with the total accumulations)?

**Authors' response**

This is the recording resolution, measurement resolution is higher. Following the referee comment to make it more clear the sentence was rearranged as follows:

„The temporal resolution of the gauges network is 1-minute. The Arpa Piemonte weather stations are equipped with CAE PMB2 tipping-bucket rain gauges. Their resolution (0.2 mm) is the amount of precipitation for one tip of the bucket. The working range of measures is from zero mm to 300 mm/h with underestimation for high precipitation intensities. Such errors are corrected according to results of WMO Field Intercomparison of Rainfall Intensity Gauges (Vuerich et al., 2009)."

**Referee's comment**

3. On lines 105-107, the computation of $\Phi_{DP}$ and $K_{DP}$ from raw $\Phi_{DP}$ is mentioned, along with the fact that "carefully tuned parameter values according to data specifics" are used for this. It would be interesting and highly relevant to include a more thorough description of this in the paper, especially since $K_{DP}$ is a key variable in this paper. I think a one- or two-sentence summary of the method would be nice, along with a short description of the parameters and how they were determined.

**Authors' response**

Added a few sentences to describe the derivation of $K_{DP}$: "With default parameter values the rays where differential propagation phase folding occurred did not unfold correctly and thus the function did not produce correct specific differential phase values. In order to fix the folding issue function parameters *self_const* (self-consistency factor) and *low_z* (low limit for reflectivity – reflectivity below this value is set to this limit) had to be tuned. The default values were 60000.0 and 10.0 respectively and after testing with various combinations of various values the values 12000.0 and 0.0 were found to produce optimal results and therefore were chosen for final calculations."

**Referee's comment**

4. On lines 107-110, the self-consistency method for re-calibrating $Z_H$ is discussed, where $Z_{DR}$ is also used. Later, on lines 124-125, it is mentioned that the use of $Z_{DR}$ for quantitative precipitation estimation is not recommended for C-band radars. I think it should be discussed here why $Z_{DR}$ can be used for re-calibration of $Z_H$.

**Authors' response**

Agreed. Following the referee comment the following sentence was added to Section 2.2:

„$Z_{DR}$ is not suitable for QPE on C-band radars, but it can be used in this calibration methodology after applying strict restrictions on the data used for this purpose."

**Referee's comment**

5. On line 127, the threshold for switching between $R(Z_H)$ and $R(K_{DP})$ is defined to be 25 dBZ for $Z_H$. Using Eqs (1) and (2), this threshold translates to R ≈ 1 mm h−1 and $K_{DP}$ ≈ 0.015° km−1. These values are much lower than what is cited from the literature (R = 50 mm h−1 and $K_{DP}$ = 0.5 – 1° km−1 ). What is the reason for using this much lower threshold? I think this should be explained in the paper.

**Authors' response**

We agree that an explanation would be suitable in the paper. Various thresholds were tested on our data and this performed the best. Following the referee comment the following sentence was added to the manuscript:

„The $Z_H$ threshold value was selected after testing with various reflectivity levels. The threshold level is considerably lower than some of the thresholds used in the literature but on our datasets it performed the best."

**Referee's comment**

6. In Section 2.2, there is no mention of attenuation that could affect the $R(Z_H)$ estimates. This attenuation could be corrected for using $\Phi_{DP}$. Is there a specific reason why attenuation correction is not carried out?

**Authors' response**

We agree that it should be mentioned in Section 2.2 and explained why it is not used. Following the referee comment the following was added to the manuscript:

"The QPE of $R(Z_H)$ can be affected by attenuation on C-band radars especially in heavy precipitation and at long distances. While this can be corrected using $\phi_{DP}$ in our study it was not applied to the reflectivity data in order to not introduce another possible source of error between the results of Estonia and Italy that could not be easily quantified. Effectiveness of attenuation correction using $\phi_{DP}$ is hampered by its temperature, shape and size distribution dependence which affect the accompanying error (Vulpiani et al., 2008)."

**Referee's comment**

7. In Section 2.2, it would be good to mention that the effect of VPR will be limited in the analyses because only data from the warm season will be used, and that only data close to the radars (70 km and 30 km for Estonia and Italy, respectively) will be used.

**Authors' response**

Agreed. Following the referee comment the following sentence was added to Section 2.2:

"The QPE of $R(Z_H)$ can also be affected by the effect of non-uniform vertical profile of reflectivity (VPR). In the current study the effect of VPR will be limited because only data from warm season was used and distance limits to the radar data were set (70 km for Estonia and 30 km for Italy, respectively)."

**Referee's comment**

8. In Section 2.2, it is not explicitly mentioned how precipitation accumulations are computed. I assume (also based on the rest of the paper) that they are computed by simply adding subsequent instantaneous radar QPE values, without any space-time interpolation. It would be good to mention that here explicitly.

**Authors' response**

Agreed. The clarification seemed to suit better to Section 2.3 where it was added to the earlier description of accumulation:

„Radar-based QPEs have been accumulated to 1-hour duration and longer durations have been calculated based on these accumulations. Accumulations were calculated by adding subsequent instantaneous radar QPE values without any space-time interpolation."

**Referee's comment**

9. Is my interpretation of Fig. 1 correct if I say that in Estonia only a circular area around the radar is used (up to 70 km range), while in Italy a rectangular area (60 × 60 km2 ) around the radar is used? If this is correct, is there an explanation of why two different areas have been used? This should be included in the paper.

**Authors' response**

We agree with the referee and thus the following explanation was added to the manuscript:

"As can be seen from Fig. 1 circular area around the radar is used in Estonia but in Italy rectangular area is used. The reason for this is that Orography in Piemonte is very complex ranging from flat plains in the Po valley (about 100 m a.s.l.) to the Alps up to more than 4,000 m a.s.l. The Bric della Croce weather radar is located on Torino hill that is about 30 km from the Alps. Therefore, the elegant and simple limitation in range by some kilometers from the radar site does not work. To avoid mountainous areas, where partial and total beam-blocking, heavy ground contamination increases, a rectangle area, that extends towards flat grounds, has been preferred."

**Referee's comment**

10. Equation (3) for Pearson's correlation coefficient is incorrect. It should be: CC = Xn i=1 (ri − rm) (gi − gm) vuutXn i=1 (ri − rm) 2 vuutXn i=1 (gi − gm) 2 .

**Authors' response**

We would like to thank the referee for pointing that out. The Equation (3) was corrected accordingly in the manuscript.

**Referee's comment**

11. For the definition of the normalized mean error in Eq. (4), the multiplication with 100% needs to be omitted in order to make it consistent with the results presented in Tables 1-6. I would also like to suggest renaming this statistic to the "normalized mean absolute error", which in my view is closer to what it actually is.

**Authors' response**

We would like to thank the referee for pointing the error out and we agree with the suggestion of renaming the statistic. Manuscript was edited according to the suggestions.

**Referee's comment**

12. The authors could consider to also normalize the RMSE in Eq. (6) with the mean gauge rainfall. In this way, all statistics will be dimensionless. This is of course just a choice, and I would also be perfectly fine with leaving the definition as it is.

**Authors' response**

We thank the referee for the suggestion but decided to leave the definition as it is.

**Referee's comment**

13. On lines 192-193, the cause for the more severe underestimation of R from $Z_H$ in Italy than in Estonia is said to be the fact that there is more intense precipitation. However, doesn't this mean that the employed Z − R relation is not suitable? Differences in raindrop size distribution (DSD)

climatologies between Estonia and Italy may also cause differences. So it would be good to comment here on the suitability of the retrieval relations (Eqs (1) and (2)) for both regions.

**Authors' response**

We agree that the retrieval relations are definitely a cause for differences among the two regions. The rationale behind using the same relations for both regions was the fact that rainfall retrieval relations always entail errors with them anyway and we wanted to keep the comparison as straightforward and homogeneous as possible. Following the referee comment the following sentences were added to Section 3.1:

„Another cause of differences between the two countries might be differences in the drop size distribution climatologies. Rainfall retrieval relations also entail errors and to keep the comparison as uniform as possible we decided to use the same relations for both Italy and Estonia."

**Referee's comment**

14. On lines 198-199, it is stated that using different time intervals can help in understanding the effect of temporal sampling differences between radar and gauges. While this is certainly true, it should also be noted that using longer accumulation intervals will also lead to less severe errors (compensating underestimates and overestimates; the $R(K_{DP})$ curve in Fig. 3 is a good example of this). I think a remark about this should also be added to the text. The same holds for line 219.

**Authors' response**

Agreed. Following the referee comment the following sentence was added to Section 3.3:

„Using longer accumulation intervals leads to less severe errors as the longer period compensates for both underestimates and overestimates."

**Referee's comment**

15. On lines 215-217, an important statement is made about the improvement that $R(Z_H, K_{DP})$ gives over the other methods. At first reading, I thought that this statement is too bold given the results presented in Table 1, but on second thought it is correct. What would have helped me is if something along the lines of "(i.e., each statistic is approximately as good at the best of the other two)" after "...other product's weak points" would have been included. You could consider including this here.

**Authors' response**

We agree that the explanation should be improved. Reworded the sentence as follows:

"$R(Z_H, K_{DP})$ shows considerable improvement by combining strong aspects of the two methods"

**Referee's comment**

16. In Fig. 5, it is interesting to see that of the 4 highest 1-hour accumulations measured by a gauge, 3 of them have significantly higher radar estimates for $R(Z_H, K_{DP})$ than either $R(Z_H)$ or $R(K_{DP})$. This means that for $R(Z_H, K_{DP})$, probably the best estimator of R is selected for most of the intervals (i.e., for at least one of the underlying 5-minute intervals $R(Z_H)$ is higher than $R(K_{DP})$, and it is correctly selected for $R(Z_H, K_{DP})$). I think this merits some more discussion in the paper, especially since this is the case for 3 of the 4 highest 1-hour accumulations.

**Authors' response**

We agree that pointing this out together with additional explanation would be useful. Following text was added to the manuscript:

"Although from Fig. 5 it can be noticed that of the four highest 1-hour accumulations measured by the gauge, three of them have significantly higher radar estimates for $R(Z_H,K_{DP})$ than either $R(Z_H)$ or $R(K_{DP})$. This could be explained by precipitation that was very variable in intensity and also in spatial coverage in these three cases which in turn caused unsteady behaviour of the precipitation estimates. $Z_H$ underestimates high intensities, but with low intensities $K_{DP}$ becomes noisy and the rainfall intensity estimation is not feasible. Finally, to reduce $K_{DP}$ uncertainties range averaging is mandatory, leading to underestimation in case of very localized showers. By blending both $R(Z_H)$ and $R(K_{DP})$, a better rainfall estimation is expected."

**Referee's comment**

17. On lines 242-243, it is stated that the normalized bias is much smaller for the 24- hour accumulations than for the 1-hour accumulations. However, looking at the definition of the NMB in Eq. (5), there should be absolutely no difference between the two, if the same underlying samples have been used (i.e., it makes no difference whether you first sum over 24 hours, and then subtract gauge from radar sums, or if you compute the difference first and then sum over 24 hours because subtraction is a linear operation). So what is the cause of these differences? Is it because you use different underlying samples, possibly by taking only accumulations above 0.1 mm (see captions of Figs 4 to 7)? If this is the case, this stresses the importance of low-intensity rain for total rainfall accumulations. This should be explained clearly. The same holds for differences between 24-hour and 1-month accumulations.

**Authors' response**

The cause is most probably different underlying samples. The 0.1 mm threshold is applied after the accumulation as a last step before calculating the verification metrics. This means that the total accumulated precipitation sum is larger in 24h accumulation dataset than in 1h dataset (although the difference is not big, 10900 mm vs 10200 mm in case of Estonia and gauge measurements). Following the referee comment the following was added to the manuscript:

"By looking at the definition of NMB in Eq. (5) it can be seen that in case the same underlying samples are used NMB should be equal on all accumulation lengths. In our study the underlying samples were different as the 0.1 mm threshold was applied after the accumulation as a last step before calculating the verification metrics. This emphasizes the importance of low-intensity precipitation for total accumulations."

**Referee's comment**

18. If I compare the NMB presented for $R(Z_H)$ in Table 3 and the corresponding panel in Fig. 6, I'm surprized at the fact that the underestimation by the radar is so small. Is this because there is an extremely high density of points just above the black line close to 0 mm in Fig. 6?

**Authors' response**

Rechecking the dataset and recalculating the NMB gave the same results so the reason behind it must be high density of points above and near the black line on low accumulation values.

**Referee's comment**

19. Comparison of Figs 5 and 7 gives me the feeling that there may be an error in one of them. For example, if I roughly add all of the accumulations from R($Z_H$) in Fig. 5, the resulting amount of rain is

much smaller than when I roughly add all of the accumulations from R(Z$_H$) in Fig. 7. Furthermore, the number of accumulations exceeding 0.1 mm given in the caption is higher for Fig. 7 than for Fig. 5. This is impossible unless a different dataset has been used. So I suggest to take another careful look at the figures and the results presented in the tables.

**Authors' response**

The comparison between radar and raingauge is carried out only if both the gauge measurement and the radar estimation are simultaneously greater than 0.1 mm; otherwise, this comparison loses meaning. Here, short duration and scattered precipitations are considered (i.e. few rain gauges record rainfall during an event). When the rainfall accumulation interval decreases, the number of valid couples (i.e. both greater than zero) tends to decrease. That's the reason because the number of samples is greater in Figure 7 than in Figure 5. As all invalid couples are discharged, it has no meaning to compare the total accumulation between these plots.

**Referee's comment**

20. Figure 7 seems to show two regimes for R(Z$_H$), where one overestimates and the other underestimates for higher rainfall accumulations. It would be interesting to discuss this in the paper. I'm interested to learn if these regimes are separable by some other variable such as time, temperature, or something else.

**Authors' response**

The reviewer is right. The Bric della Croce weather radar is located on a top of hill at 770 m asl and during the winter season a vertical profile reflectivity correction (VPR) is applied (Koistinen, 1991). This correction is manually switched on at the beginning of the cold season and it is switched off at the end. In case of convective precipitation, this correction may lead to rainfall overestimation. On the other hand, stratiform cold precipitation is heavily underestimated when VPR correction is switched off. So, the VPR correction leads to these regimes in daily comparison (Figure 7). The separation between the two regimes could be obtained by reducing the study area even more and limiting the study to June, July and August. Unfortunately, the corrected reflectivity (including VPR) is available for studied years only; later both corrected and uncorrected become available. The following Figure shows the same scatterplot as in the paper but limited to summer months July and August 2012-2016.

[Figure]

The double regime induced by VPR correction disappears. However, if we consider the logarithmic ratio between rainfall estimated by weather radar and measured by gauge ($af = 10\ log10(R/G)$), it

is clearly visible a seasonality with underestimation (on average) for April, May and October and November and slight overestimation during warm months.

[Figure]

Cited references:

Koistinen, J.: Operational correction of radar rainfall errors due to the vertical reflectivity profile, in: Proceedings of the 25th Radar Meteorology Conference, American Meteorological Society, Paris, France, 91–96, 1991.

Vuerich, E., Monesi, C., Lanza, L., Stagi, L., Lanzinger, E.: WMO Field Intercomparison of Rainfall Intensity Gauges, Vigna di Valle, Italy, October 2007 - April 2009, WMO/TD- No. 1504; IOM Report-No. 99, available at: https://www.wmo.int/pages/prog/www/IMOP/publications/IOM-84_Lab_RI/IOM-84_DataSheets/TippingBucket_Italy_CAE.pdf, 2009.

Vulpiani, G., Tabary, P., Parent du Chatelet, J. and Marzano, F.S.: Comparison of advanced radar polarimetric techniques for operational attenuation correction at C band, J. Atmos. Ocean. Technol., *25*, 1118-1135, https://doi.org/10.1175/2007JTECHA936.1, 2008.

---

## Author Response (AR2)

**Responses to comments from Anonymus Referee #1**

On revised version of „Applicability of dual-polarization weather radar quantitative precipitation estimation for climatological purposes" by Tanel Voormansik et al.(HESS-2019-624)

GENERAL COMMENTS

**Referee's comment**

The paper has been revised by the authors and some additional explanations and clarifications concerning the data and the processing methods have been brought. Some references have been added. This is highly appreciated. I would like to thank the authors for the careful attention given to most specific comments.

However, some majour concerns I expressed in my review remain in the revised version and, unfortunately, some suggestions for improving the paper have not been considered by the authors. The following issues remain :

- the focus of the paper is still unclear. The title has been changed but the focus is still on the use for climatological purpose while the scope of the paper is on the evaluation of the performance of the QPE methods.

**Authors' response**

Authors would like to sincerely thank the referee for the time and effort spent in reading the improved manuscript and for making many clear and constructive suggestions for further improvement. These suggestions were taken into account and the manuscript was edited accordingly. The title has been rephrased to fit more with the scope of the paper. The new proposed title is "Evaluation of the dual-polarization weather radar quantitative precipitation estimation using long-term datasets".

**Referee's comment**

- the tuning that was applied to optimize the processing is still unclear. The settings have been selected in order to produce the optimal results but very little is said on how the evaluation of the quality of the results has been performed in order to find the best parameters.

**Authors' response**

We agree that the manuscript would benefit from providing more details about the evaluation of the quality of the results in the section where the derivation of KDP is discussed. Following the referee comment, the manuscript was improved by adding details about the evaluation process. The detailed answer regarding the processing and evaluation of these results is provided under the specific comments section below.

**Referee's comment**

- the impact of re-calibrating the horizontal reflectivity has not been analyzed as suggested in the review.

**Authors' response**

Following the referee comment the manuscript was improved further by adding the description of the impact of re-calibration as the referee suggested:

"The impact of the re-calibration was evaluated on one month of 1-hour accumulation data from August 2018 using the verification measures introduced in Sect. 2.1 (Eq. 1-5). The verification results are presented in Table 1. QPE product based on re-calibrated reflectivity ($R(Z_{H\ cal})$) shows clearly superior results compared to the non-calibrated reflectivity based product ($R(Z_{H\ def})$), most notably by decreasing the negative bias."

**Table 1.** Verification results of the test dataset of one month (August 2018) of the radar-based rainfall 1-hour accumulation products of Estonia.

| | $R(Z_{H\ cal})$ | $R(Z_{H\ def})$ | $R(K_{DP\ tuned})$ | $R(K_{DP\ def})$ | $R(Z_{H15}, K_{DP})$ | $R(Z_{H20}, K_{DP})$ | $R(Z_{H25}, K_{DP})$ | $R(Z_{H30}, K_{DP})$ | $R(Z_{H35}, K_{DP})$ |
|---|---|---|---|---|---|---|---|---|---|
| CC | 0.699 | 0.699 | 0.659 | 0.428 | 0.687 | 0.721 | 0.726 | 0.713 | 0.705 |
| NMAE | 0.572 | 0.634 | 1.074 | 1.491 | 0.855 | 0.69 | 0.605 | 0.596 | 0.595 |
| NMB | -0.212 | -0.421 | 2.652 | 4.958 | 1.441 | 0.655 | 0.067 | -0.118 | -0.184 |
| RMSE (mm) | 1.611 | 1.709 | 2.329 | 2.656 | 2.071 | 1.832 | 1.714 | 1.718 | 1.704 |
| NASH | 0.247 | 0.202 | -0.088 | -0.241 | 0.032 | 0.144 | 0.199 | 0.197 | 0.204 |

**Referee's comment**

- using the Italian data, it was possible to analyse the impact of the temporal sampling, 5-min versus 15-min. A simple method was proposed that would have allowed to evaluate errors resulting from temporal sampling.

**Authors' response**

[Figure]

The Figure shows the effect of time sampling on QPEs. The scatterplot shows 1-hour precipitation accumulation reflectivity-based QPE, derived by 5-minutes sampling versus 15-minutes sampling. QPEs have derived from the second elevation PPI of Bric della Croce weather radar. The comparison is derived from hourly accumulation rainfall observed on October $10^{th}$, 2020 and the sample size is 253,514. As expected, the normalized mean bias is close to zero, while the **cross-correlation** is **0.922** and **the normalized mean absolute error is 0.21**.

If we compare different skill scores for 1-hour QPEs in Estonia and Italy, part of differences in cross-correlation and normalized mean absolute error can be explained as due to different time sampling. Table 2 below summarizes cross-correlation and normalized mean absolute error in Estonia and Italy.

**Table 2.** Verification of the 1-hour accumulation QPE products of Estonia and Italy and differences without ("Difference") and with ("Comp. Diff") compensating the impact of the temporal sampling. CC and NMAE values are obtained from Table 3 and Table 4.

| | $R(Z_H)$ | $R(K_{DP})$ | $R(Z_H,K_{DP})$ | | $R(Z_H)$ | $R(K_{DP})$ | $R(Z_H,K_{DP})$ |
|---|---|---|---|---|---|---|---|
| CC (Estonia) | 0.679 | 0.674 | 0.697 | NMAE (Estonia) | 0.537 | 0.868 | 0.594 |
| CC (Italy) | 0.843 | 0.808 | 0.870 | NMAE (Italy) | 0.531 | 0.514 | 0.423 |
| Difference | -0.164 | -0.134 | -0.173 | Difference | 0.006 | 0.354 | 0.171 |
| Comp. Diff. | -0.086 | -0.056 | -0.095 | Comp. Diff. | -0.204 | 0.144 | -0.039 |

Compensating the values obtained in Estonia for loss of correlation (0.078) and increased NMAE (0.21) due to 15-minutes time sampling with values estimated in Italy, it is visible that CC and NMAE are comparable in Estonia and Italy (last row in Table 2). It is worth noting that after the compensation is applied, QPE estimated by $R(Z_H)$ shows lower NMAE in Estonia. The difference in NMAE of $R(K_{DP})$ and $R(Z_H)$ QPEs might stem from different precipitation regimes (more intense precipitation in Italy).

**Referee's comment**

- the original aspect of the present study does not clearly appear
The paper has been improved but, in my view, only minor modifications have been brought by the authors.

**Authors' response**

Following the referee's comment, the manuscript was improved further to make the original aspects of the study stand out more clearly.

SPECIFIC COMMENTS

Abstract

**Referee's comment**

The length of the dataset is now mentioned. It is 5 years for both radars. It remains very short for climatological applications.

**Authors' response**

We agree that in a traditional meaning 5 years is very short to use the term "climatology". Therefore we changed the title of the paper and used the term "long-term" instead. Nevertheless, it has to be pointed out that in a number of studies with similar dataset lengths the term "climatology" has been used before. For example Kaltenboeck and Steinheimer (2015) used a 5 years of convective seasons in their paper titled "Radar-based severe storm climatology for Austrian complex orography related to vertical wind shear and atmospheric instability"

Chen et al (2012) in their study "Diurnal variations in convective storm activity over contiguous North China during the warm season based on radar mosaic climatology" (https://doi.org/10.1029/2012JD018158) used 4 years of warm season data.

"In radar climatology, the large spatial coverage and resolution of radar observations might allow us to partly make up for short time series by "trading space for time," that is, by retrieving larger samples of relevant events under the—admittedly strong—assumption that these can be considered as independent." Saltikoff et al (2019), An Overview of Using Weather Radar for Climatological Studies: Successes, Challenges, and Potential https://doi.org/10.1175/BAMS-D-18-0166.1

**Referee's comment**

1. Introduction

The aim of the study is described like this in the introduction :
"The main aim of this study is to evaluate the potential of using polarimetric weather radar QPE for climatological evaluation of precipitation regimes."
What do the authors mean with "precipitation regimes" ?

**Authors' response**

We agree that this sentence would benefit from better formulation and we rephrased it as follows: "The main aim of this study is to evaluate the potential of using polarimetric weather radar QPE on long-term warm-season datasets in various climatological environments."

**Referee's comment**

2. Data and methods

2.1 Rain gauge measurements

A short description has been added. Can the authors explain which kind of "weather sensor data" they use in the manual quality control ?

**Authors' response**

Weather sensor data used for manual quality control are obtained from Vaisala Present Weather Detectors PWD52 and PWD22 (precipitation type, intensity and accumulation) located on the same stations.

**Referee's comment**

2.2 Weather radar precipitation estimates

Some additional information is provided in the revised version concerning the processing of the raw PHIDP data to derive KDP. However, some questions remain. For example, what is the resolution in range of the derived KDP ? Based on which criteria are the results considered as optimal ?

**Authors' response**

The results were considered optimal after comparisons of the resulting KDP rainfall fields and verification statistics of a test period. The values were first chosen after preliminary tests with single scans from multiple years between 2011-2018 and then confirmed after a final test with one month of 1-hour accumulation data from August 2018. The quality of the results was evaluated by using the verification measures introduced in Sect. 2.1 (Eq. 3-7).
The KDP retrieval process involves filtering that reduces the range resolution of KDP to approximately 1 km.

[Figure]

KDP rainfall fields. On the left based on default parameters of the Py-ART phase_proc_lp function and on the right based on the optimal parameter values that were chosen for final calculations (parameters self_const (self-consistency factor) and low_z (low limit for reflectivity – reflectivity below this value is set to this limit) values were changed from 60000.0 and 10.0 to 12000.0 and 0.0 respectively. It is visible that with the default parameter values (left) the rays where differential propagation phase folding occurred did not unfold correctly and thus the function did not produce correct specific differential phase values. Finding the optimal parameters fixed that issue (right).

Following the referee comment the following was added to the manuscript:
"The values were first chosen after preliminary tests with single scans from multiple years between 2011-2018 and then confirmed after a final test with one month of 1-hour accumulation data from August 2018. The quality of the results was evaluated by using the verification measures introduced in Sect. 2.1 (Eq. 1-5). The final test results are shown in Table 1. The product with optimal values for the KDP processing algorithm ($R(K_{DP\ tuned})$) improves all verification measures when compared to the product based on the KDP processing with default parameters values ($R(K_{DP\ def})$). The KDP retrieval process involves filtering that reduces the range resolution of KDP to approximately 1 km."

**Referee's comment**

The selection criteria based on PHIDP and Delta PHIDP are not perfectly clear (L 142 to 148). It is indicated that bins with PHIDP larger than 12 deg. are removed and that ΔPHIDPobs should be larger than 8 deg. Is this not too restrictive ?
How is the 25 dBZ threshold selected ? The authors explain that this threshold is chosen because it is the one which performs best without explaining in the revised version how this performance is evaluated. Which time period is used to make this evaluation ?

**Authors' response**

The reason to use the upper PHIDP limit of 12 degrees is that Zh and ZDR can be reduced due to attenuation in heavy precipitation. The reason why to use the lower PHIDP limit of 8 degrees is that PHIDPobs is noisy in light rain.

We agree that the selection of the 25 dBZ threshold would benefit from additional details. To improve the manuscript, the following was added to the paper:
"The reflectivity threshold was selected after verifying QPE performances at different reflectivity levels from 15 dBZ to 35 dBZ by 5 dBZ steps. The evaluation is based on 1-hour accumulation rainfall for August 2018 in Estonia and the verification statistics introduced in Sect. 2.1 (Eq. 1-5) are applied using the same gauges employed in the latter parts of the study as reference. From Table 1 it can be seen that best scores are reached using 25 dBZ (QPE product $R(Z_{H25},K_{DP})$)."

The same evaluation for the $R(Z_H,K_{DP})$ algorithm was carried out for Bric della Croce in Italy. The Table below shows skill scores for hourly rainfall accumulation for convective precipitation on June 9th, 2020 derived comparing QPEs based on traditional $Z_H$ and $R(Z_H,K_{DP})$ with 72 automatic rain gauges, located within 50 km from the radar site. One-hour accumulations QPEs verification for different reflectivity thresholds for $R(Z_H,K_{DP})$ demonstrates that 25 dBZ threshold gives best performances.

| | $R(Z_{H15}, K_{DP})$ | $R(Z_{H20}, K_{DP})$ | $R(Z_{H25}, K_{DP})$ | $R(Z_{H30}, K_{DP})$ | $R(Z_H)$ |
|---|---|---|---|---|---|
| CC | 0.864 | 0.862 | 0.865 | 0.869 | 0.778 |
| NMAE | 0.39 | 0.39 | **0.38** | 0.38 | 0.42 |
| NMB | 0.08 | 0.07 | **0.01** | -0.09 | -0.26 |
| RMSE (mm) | 4.5 | 4.65 | 4.66 | 4.68 | 4.14 |
| NASH | 0.29 | 0.29 | 0.29 | 0.29 | 0.25 |

**Referee's comment**

2.3 Comparison framework

Concerning hail, the following sentence was added by the authors : "Possible occurrence of hail was not removed from the data because of the intention to keep additional data processing minimal and allow level comparison of the various QPE methods.".
I don't catch what "level comparison" means.

**Authors' response**

By "level comparison" we mean equality among the QPE methods. We intend to keep processing minimal to be able to compare the baseline performance of each method. The word "level" was replaced by "equal" in the manuscript to be more easily understandable.

**Referee's comment**

3. Results and discussion

The discussion of the results has been improved but the interpretation remains sometimes unclear.

**Authors' response**

Following the referee's comment, the Results and discussion section was improved by updating Figure 9 and Figure 10 to be more clearly understandable.

**Referee's comment**

TYPOS AND FORMULATIONS
Some strange formulations and spelling errors remain present throughout the text. I would recommend having the text proofread by a native English speaker.

**Authors' response**

Following the referee's comment, the whole manuscript was checked for formulation, spelling and grammar errors and improved accordingly.

**Responses to comments from Anonymus Referee #2**

On revised version of „Applicability of dual-polarization weather radar quantitative precipitation estimation for climatological purposes" by Tanel Voormansik et al.(HESS-2019-624)

GENERAL COMMENTS

**Referee's comment**

The Manuscript deal with a relatively long period of 5 summer seasons of two weather radar observations in Italy and Estonia. The motivation of the work is convincing to me although 5-seasons cannot be properly defined as climatological dataset. This must be stressed through the manuscript. The results are not really new as implicitly recognised by the Authors. However, the relatively long time period and sites analysed are worth to be considered for a possibile publications.

I am in doubt about the appropriateness of the journal with respect the novelty of the work presented.

**Authors' response**

Authors would like to sincerely thank the referee for the time and effort spent in reading the improved manuscript and for making a number of constructive suggestions for improvement. This helped a lot to improve the manuscript. We agree that in a traditional meaning 5 years is short to use the term „climatology". Therefore we changed the title of the paper and used the term "long-term" instead. The originality of the study is ensured mostly by the length of the dataset that was used. There have been several studies using selected short cases or datasets consisting of a number of days. To our knowledge, this is the first time multi-year dual-polarization radar QPE methods are studied as is stressed also in the manuscript.

SPECIFIC COMMENTS

**Referee's comment**

- Since the manuscript focuses on summer seasons I guess that convective events mainly populate the analysed dataset. The representativeness of the dataset considered is then biased toward more severe events. So, maybe the manuscript's title is not fully appropriate. Please consider this option "Applicability of dual-polarization weather radar quantitative rainfall estimation for climatological extremes"

**Authors' response**

We agree with the comment as the convective events are quite frequent in the warm season. To make it more clear for the reader we improved the introduction to put more emphasis on the use of warm-season data and also changed the title to be more appropriate: "Evaluation of the dual-polarization weather radar quantitative precipitation estimation using long-term datasets". Because we used all the data from the period which includes also stratiform precipitation and to show that we are not picking only severe cases we would not use the term "extremes".

**Referee's comment**

- Path attenuation in convection can be severe at C band but is not corrected by the authors to avoid introducing further errors. On the other hand, Pdp (Kdp actually) is used for QPE. So If you trust in Kdp for QPE you should do the same for path attenuation compensation too. Isn't it?

**Authors' response**

Adding path attenuation correction carries along risks that were also elaborated more in the last version of the manuscript. Its performance is dependent on multiple external parameters (temperature, drop shape and size distribution) which makes it difficult to quantify the error in various situations and on both locations, Estonia and Italy. Path attenuation might also fail in hail. What is more, we intended to keep the comparison simple.

**Referee's comment**

- Conclusions of the work are not really new. See for example ( https://doi.org/10.1175/JAMC-D-10-05024.1 and https://doi.org/10.3390/atmos8020034)

**Authors' response**

We would like to thank the referee for pointing at the examples, references for both papers have been now added to the manuscript as we find them highly relevant to our study as well. We agree that the conclusions are similar, but as we have stated in the manuscript, one of the main strengths compared to earlier studies on the same topic is the length of our used datasets. The first referred example (https://doi.org/10.1175/JAMC-D-10-05024.1) is based on 10 selected days and is already cited in the manuscript. The second example (https://doi.org/10.3390/atmos8020034) is based on 3 days of data.

Minor comments:

**Referee's comment**

- Introduction L 30 "Detailed surface rainfall information is of great importance in many fields not only for agricultural or hydrological applications but also for assimilation purposes within numerical weather models and climatologies."
I do not think rain is directly assimilated into NWP. About Climatological models verify their output with large scale (100km at best) satellite retrievals . Thus the required rain fields for climatology do not need to be too much "detailed ". By reading next It is clear what do you mean but I suggest to rephrase the beginning of the manuscript.

**Authors' response**

We agree with the comments and to make the beginning of the manuscript more clear the following was added to the manuscript:
"Detailed surface rainfall information is of great importance in many fields not only for agricultural or hydrological applications. In the recent past the COST 717 Action entitled "Use of Radar Observations in Hydrological and NWP models" investigated the assimilation of weather radar based precipitation in NWP (Macpherson, 2004). Weather radar data have been assimilated in a variety of assimilation systems and models of increasing resolution. At the beginning the latent heat nudging was the most popular technique (Gregorč et al., 2000), while researchers recently moved towards volume reflectivity assimilation techniques: for example, Schraff et al. (2016) proposed the KENDA (ensemble

Kalman filter for convective-scale data assimilation) operator to assimilate reflectivity volume data in the COSMO (COnsortium for Small-scale MOdelling) model."

**Referee's comment**

- Could you please give more explicit detail on the automatic gauge quality control?

**Authors' response**

Automatic checks are performed at real-time rain gauges data collection. First of all, range controls verify that the instrumental range is correct. Then, time series checks - e.g. constant values for several hours suggest instrumental anomaly - and cross-checks with other sensors (i.e. temperature and relative humidity) are performed. Finally, proximity checks (i.e. between close rain gauges) are performed to avoid zero rainfall accumulation recorded when all neighbouring rain gauges are recording precipitation.

**Referee's comment**

- Page 5, "self_const (self-consistency factor) " is not explained.

**Authors' response**

Agreed. Short description of the factor added to the manuscript:
"Self-consistency factor takes into account the spatial variability of reflectivity and differential reflectivity within a given path. It is used to improve KDP field behaviours to more closely follow the cell patterns found in Zh."

**Referee's comment**

- please increase the readability of figure 9 and 10

**Authors' response**

We agree that the readability of Figures 9 and 10 was not sufficient. The figures were updated to increase readability. Updated figures:

[revised manuscript text omitted]